# Týr-the-Pruner: Structural Pruning LLMs via Global Sparsity Distribution Optimization

**Guanchen Li, Yixing Xu, Zeping Li, Ji Liu, Xuanwu Yin, Dong Li, Emad Barsoum**
Advanced Micro Devices, Inc. (AMD)
{guanchen,yixing.xu,zeping.li,ji.liu,xuanwu.yin,d.li,emad.barsoum}@amd.com

## Abstract

Structural pruning enhances hardware-agnostic inference efficiency for large language models (LLMs) yet often fails to maintain comparable performance. Local pruning performs efficient layer-by-layer compression but ignores global topology. Although global pruning aims to identify an optimal sparse model, intuitive methods typically adopt a two-stage paradigm that first evaluates substructure saliency and then applies global pruning, which ignores inter-structure dependencies and fails to achieve end-to-end optimization. To address these limitations, we propose **Týr-the-Pruner**, an efficient end-to-end search-based global structural pruning framework. This framework constructs a supernet by repeatedly applying local pruning across a range of sparsity ratios to each layer in an LLM, with the core goal of determining the optimal sparsity distribution under a target overall sparsity ratio. Concretely, we introduce an effective local pruning and an expectation error accumulation approach to improve supernet construction. Furthermore, we employ an iterative prune-and-search strategy with coarse-to-fine sparsity granularity to ensure efficient search convergence. Experimental results show that Týr-the-Pruner achieves state-of-the-art structural pruning, retaining **97%** of the dense model's performance while removing a challenging **50%** of Llama-3.1-70B's parameters.

## 1 Introduction

Large language models (LLMs) have significantly advanced natural language processing, achieving exceptional performance in tasks such as text understanding, generation, and reasoning [53, 7, 3]. However, the computational and storage resources required for model deployment incur high costs and environmental impacts, limiting their accessibility in resource-constrained scenarios. Model compression techniques, such as quantization [21, 10], pruning [9, 23, 39], and low-rank decomposition [45], are essential for reducing LLM size and computational demands. This paper focuses on structural pruning, which enhances inference efficiency in a hardware-agnostic manner.

Existing structural pruning methods for LLMs are typically classified into local and global techniques. Local pruning methods [17, 26], which prune layers individually, enable efficient compression of hundred-billion-scale LLMs on a single GPU via offload approaches. However, they overlook global dependencies in model topology and restrict the sparsity to be uniform across layers. Global pruning methods [23, 18, 1] alleviate local constraints, facilitating sparsity allocation and the potential for optimal pruning. However, many existing methods estimate the saliency of local substructures and prune them accordingly via global ranking, ignoring inter-structure dependencies and hindering end-to-end optimization. Such methods may also suffer from the inefficiency of backpropagation-based saliency estimation and overfitting when calibration data is limited. Therefore, a question arises:

*How to achieve **efficient global** structural pruning with **end-to-end** optimization?*

39th Conference on Neural Information Processing Systems (NeurIPS 2025).

To address this challenge, we propose **Týr-the-Pruner**, an efficient search-based global pruning framework with end-to-end optimization. Our framework constructs a supernet by applying local pruning to each layer, producing pruned copies with different sparsity ratios. The objective is to identify an optimal subnet that satisfies the target overall sparsity ratio within the supernet by determining the optimal sparsity distribution across layers. We use evolutionary search [22] to solve this optimization problem. To construct reliable supernets and perform effective and efficient search, we make the following contributions:

- **To improve supernet construction**, we propose an effective local pruning approach for attention heads and feed-forward networks (FFN), using Taylor expansion-based first- and second-order optimization information to identify redundant structures and adjust remaining weights. Pruning and weight adjustments are applied progressively and finely to preserve accuracy. Additionally, we introduce an expectation error accumulation approach to address the challenge of unclear error propagation caused by the multiple pruned copies within the supernet. This approach ensures balanced mutual awareness across sparse structures during supernet construction.

- **To enhance the efficacy and efficiency of subnet search**, we employ a tailored distillation-inspired metric as the optimization objective to guide the search process, aiming to preserve the subnet's generative capability. In general, Týr-the-Pruner is formed as an iterative prune-and-search framework that refines sparsity allocation for each layer with reduced search space and fast convergence. Each iteration prunes and constructs a supernet across a specific range of sparsity ratios, coupled with a sparsity-shift-driven evolutionary search, where random sparsity shifts between layers generate parent candidates, and the best-performing ones are filtered as offspring. The sparsity interval is refined after each iteration.

By making these contributions, Týr-the-Pruner achieves end-to-end global pruning with strong efficacy and efficiency. Notably, the proposed framework only requires 4M tokens for calibration and search. Experimental results demonstrate that Týr-the-Pruner surpasses state-of-the-art pruning methods. For example, Týr-the-Pruner outperforms the SOTA method FLAP, achieving 3.45 lower perplexity in language comprehension and 10.26% higher average downstream accuracy when pruning 37.5% of the parameters of Llama-3.1-8B. Moreover, it maintains 97% performance with 50% pruning on Llama-3.1-70B, a sparsity ratio that is considered aggressive for existing methods.

## 2 Method

This section presents Týr-the-Pruner, a novel structural pruning framework for large language models (cf. Section 2.1 for preliminaries), as illustrated in Figure 1. This framework **(1) constructs a supernet** by applying local pruning across various sparsity ratios to each model layer, aiming to **(2) search the optimal sparsity distribution** under a target overall sparsity ratio. Specifically, we propose an effective local pruning approach (cf. Section 2.2) and an expectation error accumulation approach (cf. Section 2.3) to enhance supernet construction. An iterative prune-and-search strategy with coarse-to-fine sparsity granularity (cf. Section 2.4) ensures efficient search convergence.

### 2.1 Preliminaries

Large language models typically use the Transformer decoder architecture [42], as shown in Figure 1(a). Each Transformer layer consists of two key components: the multi-head self-attention (MHA) and the feed-forward network (FFN), followed by a residual connection and layer normalization. Given the input $\mathbf{X}_{\ell-1}$ to the $\ell$-th layer, the output $\mathbf{X}_\ell$ can be expressed as:

$$\begin{aligned} \mathbf{X} &= \text{LayerNorm}\left(\mathbf{X}_{\ell-1} + \text{MHA}(\mathbf{X}_{\ell-1})\right), \\ \mathbf{X}_\ell &= \text{LayerNorm}\left(\mathbf{X} + \text{FFN}(\mathbf{X})\right). \end{aligned} \tag{1}$$

The MHA mechanism captures dependencies across different positions in the input sequence with multiple attention heads, each with its query ($\mathbf{W}_q$), key ($\mathbf{W}_k$), value ($\mathbf{W}_v$), and out ($\mathbf{W}_o$) linear transformations. Modern LLMs typically employ a SwiGLU-based FFN [34], consisting of gate ($\mathbf{W}_{gate}$), up ($\mathbf{W}_{up}$), and down ($\mathbf{W}_{down}$) linear transformations, with activation after the gate. This structure aids in extracting non-linear representations.

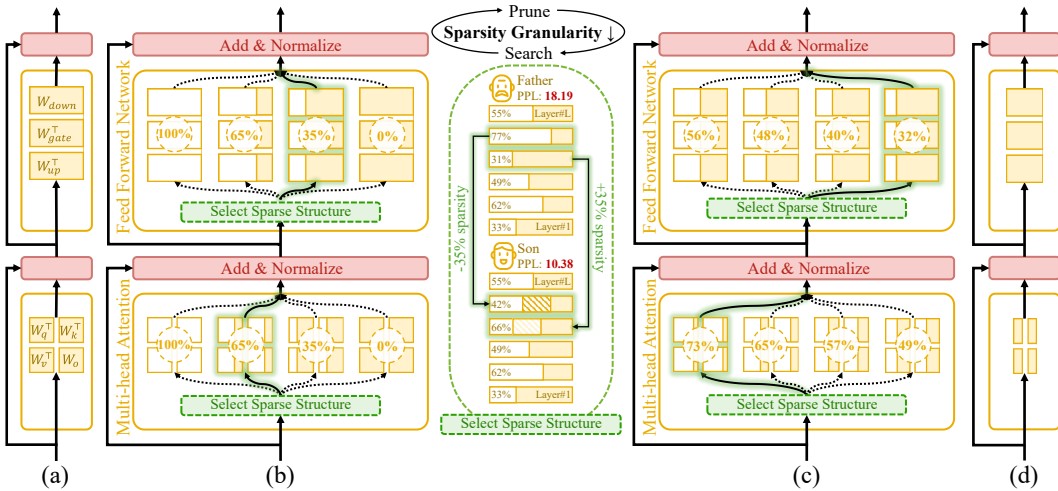

Figure 1: **An overview for Týr-the-Pruner**. Large language models (a) will be effectively locally pruned across multiple sparsity ratios and constructed into a supernet (b). An iterative prune-and-search strategy will be used to select the optimal sparse structure for each layer while maintaining a target overall sparsity ratio: pruning and sparsity-shift-driven evolutionary search are implemented iteratively with a coarse-to-fine sparsity interval granularity (c). Ultimately, the post-pruned LLM with the optimal sparsity distribution (d) is obtained.

Structural pruning for LLMs can be applied across four key dimensions: (1) attention heads, (2) FFN intermediate neurons, (3) embedding dimension size, and (4) model depth. It can be isotropic (uniform sparsity across layers) or non-isotropic (layer-specific sparsity). This paper focuses on pruning attention heads and FFN intermediate neurons with non-uniform sparsity: pruning functionally independent heads and neurons allows for controllable accuracy loss, while layer-specific sparsity further enhances pruning by tailoring compression to each layer's characteristics.

## 2.2 Effective Local Pruning

**Redundant structure identification and weight adjustment**. When pruning is scoped to the local level, one can determine the pruning outcome by eliminating the redundant input channels of each `o_proj` and `down_proj` modules, with a consistent sparsity across layers. Assuming the weight of a layer is $\mathbf{W} \in \mathbb{R}^{d_{\text{in}} \times d_{\text{out}}}$ and its input activation is $\mathbf{X} \in \mathbb{R}^{d_{\text{N}} \times d_{\text{in}}}$, the pruned weight $\widehat{\mathbf{W}}$ satisfies the sparsity constraint $C$. The corresponding optimization objective is expressed as:

$$\text{argmin}_{\widehat{\mathbf{W}}} \, ||\mathbf{X}\mathbf{W} - \mathbf{X}\widehat{\mathbf{W}}||_2^2 \quad \text{s.t.} \quad \mathcal{C}(\widehat{\mathbf{W}}) = C. \tag{2}$$

The pruning process can be viewed as a perturbation applied to the weights: $\widehat{\mathbf{W}} = \mathbf{W} - \delta\mathbf{W}$. Therefore, the error function is given by $E = ||\mathbf{X}\mathbf{W} - \mathbf{X}\widehat{\mathbf{W}}||_2^2 = ||\mathbf{X}\delta\mathbf{W}||_2^2$, which can be approximated by a Taylor series expansion around $\mathbf{W}$ and whose local fluctuations can be defined as:

$$\delta E = \underbrace{\left(\frac{\partial E}{\partial \mathbf{W}}\right)^\top \delta\mathbf{W}}_{\mathbf{G}^\top \not\approx 0} + \frac{1}{2}\delta\mathbf{W}^\top \underbrace{\frac{\partial^2 E}{\partial \mathbf{W}^2}}_{\mathbf{H} \neq 0} \delta\mathbf{W} + \underbrace{O\left(||\delta\mathbf{W}||^3\right)}_{\approx 0}. \tag{3}$$

$\delta E$ reflects the effect of $\delta\mathbf{W}$ on the pruning error, which we aim to minimize. The first-order gradient $\mathbf{G}$ cannot be neglected, as the calibration samples are inevitably misaligned with the proprietary closed-source pre-training data. The Hessian matrix $\mathbf{H}$ helps to identify pruning-sensitive weights from a curvature perspective. Considering the sparsity constraint ($\delta\mathbf{W}_{p,:} = \mathbf{W}_{p,:}$ the $p$-th input channel of $\mathbf{W}$ is to be pruned), we design the redundant channels and weight adjustment as follows:

$$\mathbf{W}_{p,:} = \mathrm{argmin}_{\mathbf{W}_{p,:}} \left( \left| \mathbf{G}_{p,:} \mathbf{W}_{p,:}^{\top} \right| + \frac{\|\mathbf{W}_{p,:}\|_2^2}{2 \left[\mathbf{H}^{-1}\right]_{p,p}} \right), \quad \delta \mathbf{W}_{\sim p,:} = -\mathbf{H}_{\sim p,\sim p}^{-1} \mathbf{G}_{\sim p,:}. \qquad (4)$$

$\mathbf{H} = \mathbf{X}^{\top}\mathbf{X}$ and $\mathbf{G} = \mathbf{H}\mathbf{W}$ (analytic solutions computed without backpropagation, efficient) are used as estimates of the local optimization information. The channel $p$ with the least error impact is identified and pruned, while $\delta\mathbf{W}$ adjusts the remaining weights to compensate for pruning errors ($\sim p$ represents other channels that have not been pruned).

**Pruning heads and neurons**. In our framework, feed-forward network neurons are pruned based on individual channel saliency computed from the `down_proj` layer, where each channel acts as the atomic unit for ranking and removal. For multi-head self-attention, saliency is first computed per output channel of the `o_proj` layer, then aggregated (averaged) across channels belonging to the same head, which is treated as the atomic unit for pruning.

**Progressively pruning and weight adjustment**. We adopt progressive pruning with an appropriately fine granularity: finer granularity enables unpruned weights to gradually and uniformly compensate for pruning losses in small increments while enabling precise and dynamic redundant channel identification. Reducing granularity does not significantly complicate pruning, as the key intermediate variable $\mathbf{H}^{-1}$ can be rapidly adjusted to account for partial channel pruning in $O(d_{\mathrm{in}}^2)$ complexity [8].

Detailed analysis can be found in Section A.1.

### 2.3 Prune-to-supernet across Multiple Sparsity Ratios

As illustrated in Figure 1(b), a supernet will be constructed by repeatedly applying local pruning across a range of sparsity ratios to each LLM layer, producing pruned copies with varying sparsity ratios. However, this introduces challenges in error accumulation across layers. Error accumulation introduces an additional forward pass of the post-pruned layer, using its output activation as input for the next layer. The change in the input directly affects the optimization of the subsequent layer. In the example shown in Figure 2, pruning half of Llama-3.1-8B's parameters using the local pruning approach with error accumulation results in significantly lower language comprehension perplexity than pruning without it. This performance gap highlights the critical role of error accumulation: it enables deeper layer pruning to be aware of shallower layer pruning.

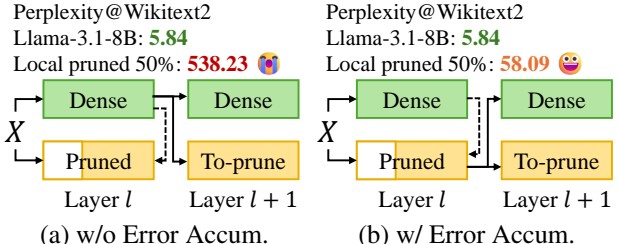

Figure 2: Implementing layerwise error accumulation gives a more accurate pruning result than not. Solid lines indicate forward propagation, and dashed lines indicate pruning.

The existence of multiple sparse structures complicates error accumulation, making it unclear which pathway to prioritize. To address this issue, we propose an *expectation error accumulation* approach to enable balanced mutual awareness among the sparse structures in the supernet. Let the output activation of the $e$-th sparse structure with sparsity $S_e$ in layer $\ell$ be $\mathbf{X}_{\ell+1,e}$. We define the expectation output activation $\mathbf{X}_{\ell+1}$ for this layer as:

$$\mathbf{X}_{\ell+1} = \sum_{e=1}^{E} \frac{1 - S_e}{\sum_{e=1}^{E}(1 - S_e)} \mathbf{X}_{\ell+1,e}. \qquad (5)$$

A higher scaling factor is assigned to low sparsity weights because their output activations are more stable and reliable. By enabling expectation error accumulation and applying the local pruning approach, we can prune Llama-3.1-8B to create nine sparse structures in each layer, with 12.5% as the sparsity interval (covers complete pruning and abandoned pruning). The post-pruned model achieves a language comprehension perplexity of 66.38 on the WikiText-2 task, with manually picking the 50% sparse structure as an example. This result is close to the ideal perplexity of 58.09 achieved under full error accumulation and significantly better than the 208.92 perplexity from random error accumulation and 538.23 perplexity with abandoned error accumulation.

## 2.4 Týr-the-Pruner

By introducing effective local pruning and expectation error accumulation approaches, we can construct a supernet to tackle the global sparsity allocation problem. Specifically, we address the following issues to achieve efficient and effective sparsity allocation: (1) defining generalizable criteria for selecting a better sparse structure, (2) achieving an efficient search-based sparse structure selection while maintaining overall sparsity, and (3) handling the contradiction between fine-grained sparsity intervals and the large search space.

**Align to dense model behaviors to win**. Towards the definition of better sparse structures, we consider that large language models are designed for multi-task generalization. Thus, guiding sparse structure selection on a single task risks overfitting. To mitigate this, we adopt a distillation-inspired metric to measure the similarity between sparse and dense models. A salient similarity indicates that the current sparse structure is better aligned with the dense model, making it more suitable for selection. Specifically, let $\mathbf{h}_\ell^{\text{dense}}$ and $\mathbf{h}_{\ell,e}^{\text{sparse}}$ denote the activations of the dense and $e$-th sparse (structure) models at layer $\ell$, and $\mathbf{z}^{\text{dense}}$ and $\mathbf{z}_{\{e\}}^{\text{sparse}}$ represent the logits of the dense model and selected ($\{e\} = \{e_\ell\}_{\ell=1}^L$) sparse subnet. The optimization objective is formulated as follows:

$$\{\hat{e}\} = \text{argmin}_{\{e\}} \sum_\ell \alpha_\ell \left\| \mathbf{h}_\ell^{\text{dense}} - \mathbf{h}_{\ell,e}^{\text{sparse}} \right\|_2^2 + \beta \, \text{KL}(\mathbf{z}^{\text{dense}} || \mathbf{z}_{\{e\}}^{\text{sparse}}). \tag{6}$$

**Sparse structure selection via evolutionary search**. Evolutionary search can achieve convergence in model architecture optimization [36, 22]. Compared to intuitive router training, evolutionary search requires no additional parameters. It maintains constant overall sparsity by shifting sparsity between sparse structures from different layers, whereas router training relies on penalty terms for suboptimal soft sparsity control. Evolutionary search is efficient, as it allows the just-in-time loading (cf. Section 3.5) of sparse structures and leverages the backpropagation-free feature.

Mutation (stochastic perturbation) in our evolutionary search arises from sparsity shifts across layers (cf. *Select Sparse Structure* in Figure 1). For instance, the sparsity of the $\ell$-th layer may decrease by $s\%$, while the $\ell'$-th layer increases by $s\%$ (achieved by selecting different sparse structures). In each generation, we randomly generate such a group of sparsity distributions as candidates. Starting from the root generation, the performance of candidates is evaluated, and the best-performing ones are selected to generate new candidates for the next generation. Generations continue to be explored until the optimal sparsity distribution is found.

**Iterative prune-to-supernet and evolutionary search**. The search space for selecting sparse structures with fine-grained sparsity is enormous. For instance, constructing a supernet with a sparsity interval of 1.5625% would result in 65 sparse structures per MHA/FFN layer. For a 40-layer LLM, this would lead to over 5K sparse structures, creating a $10^{145}$-scaled search space. Identifying solutions in this large search space is difficult and costly. To address this challenge, Týr-the-Pruner adopts an iterative prune-and-search strategy that progressively refines sparsity granularity. In each iteration, we (1) prune the model, (2) construct a supernet with a narrower sparsity interval, and (3) perform evolutionary search to locate the optimal sparsity distribution. This coarse-to-fine paradigm improves both search efficiency and accuracy in sparse structure discovery, as illustrated below.

---

**Example: Týr-the-Pruner**

Týr-the-Pruner starts by constructing a coarse-grained supernet (e.g., Figure 1(b)) using a 12.5% sparsity interval, where each MHA/FFN layer contains only nine candidate structures. Evolutionary search is then applied to determine the optimal sparsity pattern at this iteration. In the subsequent iteration, a new supernet (e.g., Figure 1(c)) is constructed by halving the sparsity interval to 6.25% and centering it on the current optimum (e.g., 37.5%), yielding nine refined candidates from sparsity 12.5% to 62.5%, followed by evolutionary search to update the optimal sparsity. With the sparsity interval halved at each iteration, Týr-the-Pruner narrows the search space to about $10^{76}$ per iteration and reaches a 1.5625% sparsity granularity within four iterations. This coarse-to-fine refinement enhances both efficiency and accuracy in identifying the optimal fine-grained sparsity for final model compression (e.g., Figure 1(d)).

---

The algorithmic procedures for local pruning, supernet construction, evolutionary search, and the overall Týr-the-Pruner framework are detailed in Algorithms 1 to 4 of Section A.2.

Table 1: **Post pruning performance comparison of different methods**. Language comprehension perplexity is validated on the Wikitext2 test set with a sequence length of 4096, where a lower value reflects better performance. Downstream accuracy (%, higher is better) is averaged across ARC-Easy, ARC-Challenge, BoolQ, HellaSwag, OpenbookQA, RTE, WinoGrande, and MMLU, with MMLU using a 5-shot benchmark and others a 0-shot benchmark. The best results are shown in **bold**.

| Sparsity | Method | Perplexity on Wikitext2 ↓ | | | | | | | Average Downstream Accuracy (%) ↑ | | | | | | |
| | | Llama-2 | | Llama-3.x | | | Mistral | | Llama-2 | | Llama-3.x | | | Mistral | |
| | | 7B | 13B | 2-3B | 0-8B | 1-8B | 7B-v0.3 | Nemo | 7B | 13B | 2-3B | 0-8B | 1-8B | 7B-v0.3 | Nemo |
| 0% | N/A | 5.12 | 4.57 | 7.29 | 5.76 | 5.84 | 4.95 | 5.35 | 57.96 | 62.05 | 57.01 | 64.08 | 64.77 | 63.72 | 66.24 |
| 12.5% | ShortGPT | 8.86 | 5.67 | 12.42 | 13.90 | 13.14 | 7.58 | 7.72 | 53.27 | 59.16 | 53.13 | 57.75 | 58.50 | 59.49 | 59.46 |
| | LaCO+ | 7.52 | 5.69 | 12.25 | 10.12 | 9.98 | 7.46 | 7.95 | 53.23 | 57.26 | 52.46 | 59.41 | 60.36 | 58.67 | 59.96 |
| | SliceGPT | 8.25 | 7.19 | 18.71 | 20.46 | 22.10 | 7.00 | 9.74 | 55.89 | 59.70 | 51.64 | 57.55 | 56.82 | 59.67 | 53.27 |
| | Wanda-sp | 6.24 | 6.09 | 182.24 | 86.91 | 18.46 | 6.86 | 7.27 | 55.40 | 57.41 | 38.02 | 33.95 | 47.89 | 59.44 | 56.82 |
| | LLM-Pruner | 6.11 | 5.17 | 11.14 | 8.24 | 8.26 | 6.17 | 6.79 | 53.38 | 59.78 | 46.98 | 53.96 | 54.04 | 55.26 | 58.23 |
| | ZipLM | 5.86 | 5.21 | 11.32 | 10.37 | 9.30 | 5.84 | 7.62 | 55.85 | 61.91 | 51.37 | 57.55 | 57.54 | 62.46 | 60.24 |
| | OSSCAR | 5.94 | 5.21 | 11.11 | 10.15 | 9.87 | 5.75 | 7.04 | 55.29 | 61.94 | 52.23 | 57.19 | 58.53 | 62.06 | 53.89 |
| | FLAP | 6.11 | 5.75 | 10.25 | 8.34 | 8.07 | 6.18 | 7.68 | 54.63 | 57.55 | 47.74 | 55.72 | 56.66 | 59.51 | 57.67 |
| | Týr-the-Pruner | **5.84** | **5.03** | **9.16** | **7.39** | **7.41** | **5.61** | **6.31** | **56.98** | **62.66** | **54.78** | **62.01** | **63.02** | **63.05** | **64.15** |
| 25% | ShortGPT | 23.41 | 17.94 | 1464.20 | 4836.41 | 3418.83 | 35.20 | 124.20 | 46.68 | 51.86 | 41.25 | 38.12 | 38.62 | 51.07 | 51.68 |
| | LaCO+ | 18.84 | 9.00 | 128.77 | 124.86 | 137.17 | 22.91 | 20.79 | 45.47 | 52.77 | 46.26 | 48.58 | 49.80 | 51.84 | 53.65 |
| | SliceGPT | 16.84 | 12.50 | 45.44 | 47.73 | 55.43 | 12.08 | 19.37 | 51.40 | 58.04 | 45.87 | 50.01 | 48.49 | 52.26 | 46.27 |
| | Wanda-sp | 9.21 | 19.92 | 94.12 | 48.95 | 962.72 | 17.83 | 15.34 | 49.92 | 38.17 | 33.93 | 34.53 | 32.40 | 49.13 | 41.30 |
| | LLM-Pruner | 11.56 | 7.11 | 25.14 | 18.65 | 19.35 | 10.24 | 11.81 | 44.09 | 49.56 | 39.55 | 42.36 | 40.88 | 46.32 | 45.26 |
| | ZipLM | 7.49 | 6.65 | 43.50 | 28.74 | 52.69 | 7.39 | 9.91 | 52.59 | 60.50 | 41.61 | 38.72 | 39.20 | 58.05 | 45.59 |
| | OSSCAR | **7.46** | 9.19 | 122.63 | 17.40 | 17.03 | 7.16 | 9.57 | 51.99 | 59.55 | 33.29 | 44.27 | 42.19 | 55.94 | 45.95 |
| | FLAP | 8.31 | 7.50 | 15.64 | **12.65** | 12.30 | 8.01 | 13.59 | 49.36 | 54.37 | 44.01 | 47.41 | 49.20 | 52.64 | 48.83 |
| | Týr-the-Pruner | 7.51 | **5.79** | 12.53 | 13.14 | **10.38** | **7.08** | **7.87** | **54.64** | **61.16** | **51.72** | **58.50** | **58.66** | **60.22** | **60.61** |
| 37.5% | ShortGPT | 70.96 | 52.24 | 554.88 | 5.1E+04 | 9.3E+04 | 2347.69 | 864.38 | 43.66 | 43.13 | 41.28 | 39.16 | 38.97 | 35.80 | 42.52 |
| | LaCO+ | 87.77 | 96.00 | 494.07 | 1645.83 | 1377.02 | 429.78 | 462.92 | 41.55 | 47.60 | 40.24 | 38.89 | 38.85 | 40.44 | 42.88 |
| | SliceGPT | 35.10 | 26.22 | 98.41 | 176.81 | 237.50 | 27.68 | 38.46 | 43.80 | 51.83 | 37.40 | 39.96 | 38.97 | 43.30 | 39.55 |
| | Wanda-sp | 19.97 | 34.70 | 344.17 | 2422.78 | 3627.00 | 31.85 | 74.87 | 40.45 | 35.69 | 33.08 | 30.59 | 32.56 | 38.13 | 33.59 |
| | LLM-Pruner | 37.75 | 14.96 | 161.10 | 87.93 | 70.93 | 24.90 | 32.10 | 35.96 | 40.36 | 33.26 | 32.40 | 32.53 | 37.94 | 37.42 |
| | ZipLM | 12.13 | 13.01 | 283.53 | 50.36 | 125.98 | 14.01 | 15.53 | 47.53 | 51.89 | 33.35 | 34.77 | 36.55 | 48.90 | 44.86 |
| | OSSCAR | 11.28 | 12.74 | 182.00 | 27.69 | 28.87 | 10.43 | 16.00 | 47.42 | 51.74 | 32.76 | 40.81 | 39.87 | 48.91 | 45.81 |
| | FLAP | 12.41 | 11.33 | **26.05** | 22.61 | 21.54 | 11.81 | 27.01 | 43.51 | 48.54 | 39.28 | 41.51 | 43.07 | 44.90 | 45.57 |
| | Týr-the-Pruner | **10.29** | **7.17** | 27.88 | **21.64** | **18.09** | **10.25** | **11.47** | **52.21** | **58.67** | **46.11** | **53.66** | **53.46** | **52.34** | **54.63** |
| 50% | ShortGPT | 226.40 | 187.23 | 2313.30 | 1473.71 | 1678.15 | 5532.76 | 6804.52 | 36.99 | 39.47 | 34.09 | 37.51 | 36.52 | 35.05 | 38.00 |
| | LaCO+ | 256.71 | 1129.00 | 6019.01 | 2.1E+04 | 5.4E+04 | 6019.01 | 5.9E+04 | 34.89 | 41.79 | 33.96 | 35.21 | 33.28 | 33.93 | 33.25 |
| | SliceGPT | 65.34 | 50.66 | 205.09 | 384.04 | 353.21 | 54.66 | 69.15 | 39.43 | 43.84 | 33.52 | 34.55 | 34.32 | 36.17 | 34.95 |
| | Wanda-sp | 122.28 | 47.89 | 262.92 | 187.41 | 188.47 | 91.34 | 293.59 | 32.26 | 35.82 | 32.29 | 33.86 | 32.39 | 33.59 | 32.27 |
| | LLM-Pruner | 117.40 | 53.96 | 473.50 | 302.15 | 288.32 | 74.04 | 469.93 | 31.70 | 35.17 | 30.97 | 31.63 | 31.58 | 32.64 | 32.89 |
| | ZipLM | 32.91 | 24.70 | 356.02 | 102.76 | 366.34 | 24.18 | 24.96 | 32.60 | 42.66 | 32.51 | 33.14 | 34.45 | 39.93 | 38.42 |
| | OSSCAR | 28.41 | 44.17 | 320.14 | 80.90 | 198.87 | 29.58 | 23.14 | 39.46 | 40.40 | 33.85 | 32.58 | 34.16 | 40.95 | 37.99 |
| | FLAP | 25.49 | 16.89 | 272.98 | 82.12 | 134.28 | 34.81 | 79.46 | 39.84 | 44.04 | 33.29 | 38.68 | 36.59 | 40.57 | 39.34 |
| | Týr-the-Pruner | **16.17** | **9.59** | **29.84** | **38.59** | **30.89** | **15.53** | **16.85** | **47.41** | **54.58** | **41.41** | **47.41** | **47.79** | **46.21** | **47.92** |

# 3 Experiments

## 3.1 Experimental Settings

**Models**. We conduct experiments using the widely adopted large language models Llama2, Llama3.x, and Mistral [41, 7, 15], focusing on models with over three billion parameters. The pruning targets include attention heads and FFN neurons, which are applied to the Transformer backbone. The `embed_tokens` and `lm_head` layers remain unchanged.

**Calibration**. For calibration, we consider FineWeb [31], a high-quality dataset curated from Common Crawl snapshots with rigorous deduplication and filtering. Specifically, we extract about 4M tokens (about 1k samples for a maximum input length of 4k) from its FineWeb-Edu subset to construct calibration samples, ensuring high data quality and efficiency.

**Evaluation**. We use perplexity as one evaluation metric for language comprehension performance [9], validated on the WikiText2 [27] test set. To evaluate the impact of compression across various downstream tasks, we report 0-shot accuracy on ARC [6], BoolQ [5], HellaSwag [51], OpenBookQA [28], RTE [43], and WinoGrande [33] tasks, as well as 5-shot accuracy on the MMLU [13] benchmark.

**Implementation details**. We implement Týr-the-Pruner with PyTorch [30] and leverage the HuggingFace Transformers and Datasets libraries [47] to manage models and datasets. For local pruning,

Table 2: **Post pruning performance on massive language models**. Accuracy (%, higher is better) serves as the comparison metric. MMLU employed a 5-shot benchmark, while other tasks used 0-shot benchmarks. The percentage of average accuracy maintenance after pruning was recorded, with values ≥95% highlighted in green and values <95% in red. The best results are shown in **bold**.

| Model | Sparsity | Method | Arc-C | Arc-E | BoolQ | HellaSwag | OBQA | RTE | WinoGrande | MMLU | AVG |
|---|---|---|---|---|---|---|---|---|---|---|---|
| Llama-2-70B | 0% | N/A | 54.44 | 82.74 | 83.73 | 64.77 | 37.40 | 67.87 | 77.98 | 68.79 | 67.22 (100%) |
| | 50% | SliceGPT | 38.65 | 68.39 | 69.63 | 38.40 | 25.00 | 63.54 | 67.40 | 50.20 | 52.65 (78%) |
| | | LLM-Pruner | 21.93 | 29.08 | 43.18 | 26.26 | 14.00 | 51.62 | 49.25 | 23.77 | 32.39 (48%) |
| | | ZipLM | 46.67 | 77.61 | 82.26 | 56.94 | 34.00 | 68.95 | 75.61 | 54.33 | 62.05 (92%) |
| | | OSSCAR | **48.21** | 78.37 | 81.99 | 57.00 | 32.60 | 67.15 | 76.64 | 56.05 | 62.25 (93%) |
| | | FLAP | 40.02 | 70.79 | 74.74 | 51.83 | 32.00 | 60.29 | 67.88 | 39.65 | 54.65 (81%) |
| | | Týr-the-Pruner | **48.21** | **79.12** | **83.18** | **60.04** | **35.20** | **70.76** | **78.14** | **60.58** | **64.40 (96%)** |
| Llama-3.1-70B | 0% | N/A | 60.58 | 87.29 | 85.29 | 66.50 | 37.00 | 70.04 | 79.64 | 78.72 | 70.63 (100%) |
| | 50% | SliceGPT | 32.08 | 58.00 | 63.85 | 34.02 | 20.60 | 53.43 | 56.99 | 32.60 | 43.95 (62%) |
| | | LLM-Pruner | 21.42 | 25.38 | 38.81 | 26.22 | 13.80 | 54.87 | 50.83 | 24.95 | 32.04 (45%) |
| | | ZipLM | 48.55 | 78.54 | 80.55 | 55.98 | 31.60 | 66.79 | 78.37 | 62.73 | 62.89 (89%) |
| | | OSSCAR | 48.29 | 78.62 | 81.44 | 54.69 | 32.80 | 68.23 | 77.58 | 60.38 | 62.75 (89%) |
| | | FLAP | 37.54 | 68.90 | 67.34 | 43.98 | 26.40 | 60.65 | 72.30 | 54.40 | 53.94 (76%) |
| | | Týr-the-Pruner | **56.74** | **85.40** | **85.20** | **64.07** | **36.40** | **71.48** | **78.91** | **70.29** | **68.56 (97%)** |

we iteratively prune and adjust weights by removing one attention head or 16 FFN neurons at a time. The prune-and-search process consists of 4 iterations, where the sparsity interval at the $i$-th iteration is set to $12.5\%/2^{i-1}$. In each iteration, we explore 50 generations with 128 offspring candidates per generation. The sparsity shifts of the attention or FFN layers are independent to ensure the consistency of the sparsity interval granularity. Candidate validation is performed using the distillation-inspired metric with vocabulary logits. We follow [36] to enhance validation efficiency: the 128 offspring are first validated on 2K tokens, and the top 16 are selected. These 16 survivors are then validated on 16K tokens, from which the top 4 are selected, and finally, the best one is validated and selected on 128K tokens. **To ensure a fair comparison**, we use the same FineWeb-Edu samples for calibration to reproduce the baselines. The benchmark results of the baselines may outperform their reported results due to the improved calibration sample size and data quality. All experiments for Týr-the-Pruner were conducted on 4 AMD Instinct™ MI250 (64GB) Accelerators, with models less than 13B parameters running on a single accelerator.

## 3.2 Performance

**Language comprehension and downstream task performance of post-pruned LLMs**. We applied structural pruning to various large language models using Týr-the-Pruner at overall sparsity levels of 12.5%, 25%, 37.5%, and 50%. The performance was benchmarked against state-of-the-art methods, including ShortGPT (layer pruning) [24], LaCO+ (ShortGPT with LaCO layer merging) [50], SliceGPT (embedding dimension pruning) [2], Wanda-SP [37, 1], LLM-Pruner [23], ZipLM [17], OSSCAR [26], and FLAP [1]. Table 1 summarizes the comparative results, highlighting post-pruning performance in language comprehension and downstream tasks (cf. Section A.8 for detailed results within each task).

Týr-the-Pruner demonstrates competitive performance across various sparsity ratios and LLMs. It consistently achieves state-of-the-art results at low sparsity ratios (≤25%). For instance, pruning 12.5% of Llama-3-8B's parameters yields the lowest perplexity (7.39) and the highest average downstream accuracy (62.37%), surpassing the previous advanced methods, LLM-Pruner and LaCO+, by 8.0% and 2.6%. At higher sparsities (≥37.5%), maintaining performance poses a significant challenge for existing methods, with advanced techniques like OSSCAR often exhibiting perplexities exceeding 100 and accuracies dropping below 40%. Týr-the-Pruner, by contrast, excels under these conditions. For example, at 37.5% sparsity, the pruned Mistral-Nemo model achieves a perplexity of 11.47 and an accuracy of 55.63%, substantially outperforming ZipLM and FLAP.

**Scale up to massive language models**. Structural pruning of massive language models challenges post-pruned performance and resource budgets. We incorporated a CPU offload policy into typical baseline methods to ensure a fair comparison on 70B-scale models. Table 2 compares the post-pruning performance of Llama-2-70B and Llama-3.1-70B at 50% sparsity.

Experimental results demonstrate Týr-the-Pruner's strong scalability under high sparsity for massive models. LLM-Pruner shows clear scaling limitations, maintaining only 48% accuracy when pruning

Table 3: **Inference efficiency of post-pruned LLMs with Týr-the-Pruner**. Benchmarks were conducted on a single AMD Instinct™ MI250 accelerator using PyTorch (HipBlas) for LLM inference, with input and output sequence lengths set to 2048.

| Model | Sparsity | #Params | TTFT | Decode Throughput |
|---|---|---|---|---|
| Llama-3.1-8B | 0% | 8.0B | 2.49 (1.00x) | 12.27 (1.00x) |
| | 25% | 6.1B | 1.94 (1.28x) | 14.13 (1.15x) |
| | 50% | 4.3B | 1.42 (1.75x) | 16.97 (1.38x) |
| Mistral-Nemo | 0% | 14.3B | 4.16 (1.00x) | 6.68 (1.00x) |
| | 25% | 11.0B | 3.34 (1.25x) | 7.55 (1.13x) |
| | 50% | 7.8B | 2.49 (1.67x) | 8.93 (1.34x) |

Table 4: **Ablation study on local pruning**. Wikitext2 perplexity and 0-shot accuracy on ARC-C, ARC-E, and BoolQ are reported.

| Method | Configuration | Wikitext2 | ARC-C | ARC-E | BoolQ |
|---|---|---|---|---|---|
| FLAP | - | 134.28 | 20.99 | 43.18 | 52.29 |
| Local Pruning | Default | 58.09 | 24.06 | 58.67 | 63.46 |
| | Wikitext2 Calibrated | 49.00 | 20.05 | 54.84 | 61.71 |
| | C4 Calibrated | 73.07 | 21.42 | 57.58 | 62.17 |
| | w/o progressive pruning | 63.48 | 23.38 | 56.65 | 62.17 |
| | w/o Hessian | 109.88 | 22.53 | 51.68 | 46.48 |
| | w/o Gradient | 67.31 | 25.60 | 57.83 | 62.17 |
| Local Pruning & Build Supernet | Default | 66.38 | 23.05 | 58.46 | 62.35 |
| | w/o Error Accum. | 538.23 | 21.93 | 33.54 | 40.31 |
| | w/ Random Error Accum. | 208.92 | 22.70 | 39.14 | 45.05 |
| | w/ Uniform Error Accum. | 75.10 | 23.72 | 53.03 | 60.06 |

Llama-2-70B. In contrast, Týr-the-Pruner achieves 97% accuracy maintenance when pruning Llama-3.1-70B, outperforming alternative methods.

**Inference efficiency of post-pruned LLMs**. To evaluate the efficiency gains of post-pruned LLMs, we constructed inference benchmarks summarized in Table 3. For Llama-3.1-8B, 50% sparsity reduces time to first token (TTFT, in seconds) by 43% and boosts decode throughput (tokens/s) by 38%. These results highlight pruning as a key technique for inference optimization in large language models. More detailed efficiency analysis can be found in Section A.4.

## 3.3 Ablation Study

**Prune-to-supernet**. The effectiveness of local pruning and supernet construction depends on factors such as calibration samples, the implementation of local pruning, and error accumulation. Table 4 presents ablation study evaluating these factors for pruning Llama-3.1-8B at 50% sparsity. Experimental results show that FineWeb-Edu is consistently preferred as a calibration source, emphasizing the importance of selecting high-quality calibration samples. The presence of both first- and second-order optimization information and progressive pruning significantly impacts accuracy, demonstrating their necessity. Furthermore, the proposed expectation error accumulation approach outperforms alternatives, showcasing its ability to make sparse structures mutually aware appropriately.

**Evolutionary search direction**. To assess the impact of search direction on final performance, we compare the effects of minimizing single-task losses versus our similarity-based metric when pruning 50% of Llama-3.1-8B's parameters, as shown in Table 5. Experiments show that single-task search underperforms our metric, which achieves optimal accuracy by calculating the similarity across activations from the first, median, last,

Table 5: **Ablation study on search direction**. Wikitext2 perplexity and 0-shot accuracy on ARC-C, ARC-E, BoolQ are reported.

| Search Direction | Wikitext2 | ARC-C | ARC-E | BoolQ |
|---|---|---|---|---|
| Wikitext2 Perplexity | 17.22 | 29.69 | 64.06 | 62.23 |
| Fineweb-Edu Perplexity | 31.65 | 31.06 | 64.18 | 62.17 |
| Similarity-based | 28.56 | 32.51 | 65.87 | 63.12 |
| Similarity-based Logits-only | 30.89 | 31.83 | 65.36 | 64.62 |

and logits layers, requiring 96 GB for hidden activation checkpointing. Due to this overhead, the logits-only metric was favored, maintaining strong performance with reduced resource demands.

**Effect comparison: Týr-the-Pruner vs. fine-grained search-only strategy**. Figure 3 demonstrates the advantages of Týr-the-Pruner over the search-only strategy in efficacy and efficiency in identifying the optimal 50% sparsity distribution on Llama-3.1-8B. In which the search-only strategy uses a fine-grained 3.125% sparsity interval. Experimental results show that Týr-the-Pruner achieves a similar convergence trend as the search-only strategy but with faster convergence, fewer generations, and reduced search time. Additionally, the final post-pruned model discovered by Týr-the-Pruner outperforms the search-only strategy, with an average accuracy of 47.79 compared to 43.58. Our

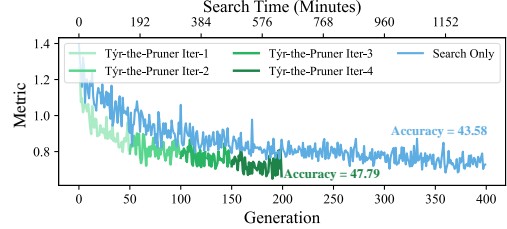

Figure 3: Týr-the-Pruner has faster convergence, fewer exploration generations, shorter search time, and better search outcomes compared to the fine-grained search-only approach.

evolutionary search maintains time efficiency, with a single generation requiring only 190 seconds.

Table 6: Týr-the-Pruner progressively refines and optimizes the sparsity distribution in iterations, steadily enhancing performance.

| Method | Wikitext2 | ARC-C | ARC-E | BoolQ | HellaSwag | OBQA | RTE | WinoGrande | MMLU | AVG |
|---|---|---|---|---|---|---|---|---|---|---|
| w/o search | 66.38 | 23.55 | 58.46 | 62.35 | 32.51 | 16.60 | 51.26 | 52.88 | 28.34 | 40.74 |
| search-only | **27.96** | 25.34 | 59.30 | 64.71 | 36.52 | 22.20 | 55.23 | 56.20 | 29.17 | 43.58 |
| Týr-the-Pruner I1 | 28.92 | 26.45 | 56.19 | 62.17 | 37.05 | 22.20 | 50.54 | 56.75 | 29.29 | 42.58 |
| Týr-the-Pruner I2 | 31.80 | 29.27 | 62.54 | 63.51 | 38.18 | 23.80 | 50.54 | 56.85 | 30.23 | 44.37 |
| Týr-the-Pruner I3 | 29.75 | 29.86 | 63.09 | 64.62 | 39.28 | 25.00 | 51.62 | 59.51 | 31.62 | 45.58 |
| Týr-the-Pruner I4 | 30.89 | 31.83 | 65.36 | 66.64 | **39.99** | 24.80 | 58.12 | **61.80** | **33.76** | 47.79 |
| Týr-the-Pruner I5 | 29.56 | **31.87** | **65.57** | **66.69** | 39.08 | **26.40** | **58.84** | 60.89 | 33.67 | **47.88** |

To further examine how Týr-the-Pruner refines model performance across successive iterations and to highlight its advantage over the search-only strategy, we report the 50% post-pruned results of Llama-3.1-8B on multiple tasks: Wikitext2 perplexity ($\downarrow$), 0-shot accuracy ($\uparrow$) on Arc, BoolQ, HellaSwag, OBQA, RTE, and WinoGrande, and 5-shot accuracy ($\uparrow$) on MMLU, as summarized in Table 6.

Experimental results underscore the superiority of Týr-the-Pruner compared to the search-only strategy. Under isotropic pruning, the w/o search baseline consistently delivers suboptimal performance across all tasks. By contrast, Týr-the-Pruner begins to surpass the search-only approach as early as the second iteration (I2), highlighting the advantage of progressively refining the sparsity distribution. The search-only method, constrained by the vast search space, suffers from extended search times and limited effectiveness. Týr-the-Pruner achieves an ideal result at the fourth iteration (I4). Although performance continues to improve at the fifth iteration (I5), the gains diminish, making four iterations the default configuration.

## 3.4 Compatibility with Quantization and Unstructured Sparsity

Týr-the-Pruner is compatible with further compression techniques. Table 7 reports results of applying quantization and unstructured sparsity to the 50% pruned Llama-3.1-8B model produced by Týr-the-Pruner. The results show that quantization preserves over 99% of the pruned model's accuracy, while unstructured sparsity at 50% and 2:4 granularity achieves performance consistent with prior work [9, 25]. These findings indicate that Týr-the-Pruner provides a strong foundation for multi-stage compression pipelines, retaining both high accuracy and robustness to further compression.

Table 7: Quantization and unstructured sparsity applied to the 50% pruned Llama-3.1-8B with Týr-the-Pruner. Reported Wikitext2 perplexity and accuracy (%) across benchmarks.

| Method | Wikitext2 | Arc-C | Arc-E | BoolQ | HellaSwag | OBQA | RTE | WinoGrande | MMLU | AVG |
|---|---|---|---|---|---|---|---|---|---|---|
| Týr-the-Pruner @ FP16 | 30.89 | 31.83 | 65.36 | 66.64 | 39.99 | 24.80 | 58.12 | 61.80 | 33.76 | 47.79 (100%) |
| AWQ [21] @ W4A16 | 34.79 | 31.06 | 64.65 | 66.09 | 39.34 | 24.60 | 60.77 | 60.77 | 31.43 | 47.34 (%99.1) |
| SmoothQuant [49] @ W8A8 | 31.31 | 31.23 | 64.94 | 65.50 | 40.18 | 25.80 | 58.84 | 61.80 | 32.45 | 47.59 (%99.6) |
| RTN @ FP8E4M3 | 31.05 | 31.31 | 65.28 | 66.45 | 40.17 | 24.60 | 58.84 | 61.48 | 32.28 | 47.55 (%99.5) |
| SparseGPT [9] @ 50% | 47.05 | 27.99 | 59.39 | 65.32 | 37.38 | 23.60 | 53.07 | 61.48 | 28.32 | 44.57 (%93.3) |
| ALPS [25] @ M4N2 | 70.22 | 23.04 | 53.62 | 63.43 | 34.22 | 21.40 | 52.71 | 58.17 | 27.34 | 41.74 (%87.3) |

## 3.5 Memory/Storage Efficiency Analysis of Týr-the-Pruner

Týr-the-Pruner employs a supernet search technique, where storing a large-scale supernet in memory is costly. To address memory concerns, we optimize our approach by storing pruned substructures on disk instead of in high-bandwidth memory (HBM). An integer Python list is used to track the currently selected substructures, en-

Table 8: Resource requirements of Týr-the-Pruner.

| Model Size | Submodules | HBM Usage | Disk Storage Usage |
|---|---|---|---|
| 7-8B | 576 | 14-16GB | 39.6GB |
| 13B | 720 | 26GB | 66.6GB |
| 70B | 1440 | 140GB | 414.7GB |

suring that only one entire LLM is loaded into HBM at any given time (e.g., the 7B model uses approximately 14GB, and the 13B model uses around 26GB). Table 8 provides detailed data on HBM and disk storage occupancy. Furthermore, since there is no dependency between iterations (iterative prune-and-search phase), the storage from previous iterations can be cleaned, further minimizing disk usage. Due to the low cost of disk storage, these memory and storage demands are highly acceptable.

# 4 Related Work

**Pruning techniques for compressing large language models**. The growing complexity of Transformer-based language models, now reaching hundreds of billions of parameters, has intensified the necessity for effective pruning strategies. Pruning methods are generally divided into unstructural and structural approaches. Unstructural pruning [9, 37] achieves high accuracy by selectively zeroing individual elements in the weight. However, it often requires specialized hardware, such as 2:4 sparse tensor cores [54], for end-to-end acceleration. Structural pruning enables hardware-agnostic acceleration by removing entire weight groups, but it may result in a pronounced loss of accuracy.

Structural pruning of LLMs can be approached as local optimization, alleviating memory constraints from loading the full model. ZipLM [17] accelerates inference by leveraging the Optimal Brain Surgeon (OBS) [12] theory, pruning weights to minimize the impact on the Hessian matrix and adjusting the remaining weights to reduce layerwise loss. Building on ZipLM, OSSCAR [26] introduces a permutation search between pruned and remaining weights within each layer, further reducing pruning-induced loss. Some approaches apply global optimization strategies to prune LLMs, overcoming local constraints, enabling customized sparsity distributions, and potentially finding optimal solutions. Fisher information was introduced as a saliency metric to guide structure pruning via global dynamic programming [18]. LLM-Pruner [23] defines broad substructure dependency groups and then evaluates their saliency to guide pruning. FLAP [1] uses a global metric that considers both weights and activations for sparsity allocation, followed by layerwise pruning and bias adjustments to mitigate pruning losses.

Additionally, there is growing interest in embedding dimension [2] and depth [24, 50] pruning techniques for LLMs. Some training-aware structural pruning methods [52, 20, 29] are also gaining attention, as they further enhance pruning effectiveness by considering training dynamics.

**Neural architecture search (NAS) for LLM compression**. Several studies have applied NAS to compress LLMs, seeking architectures that reduce inference costs while maintaining accuracy. multi-objective NAS has been employed to explore various search space definitions, identifying compressed LLM architectures that enhance efficiency and accuracy when fine-tuned on specific downstream tasks [16]. LLaMAFlex [4] fine-tunes LLMs into supernets with a Gumbel softmax-based trainable subnet router, realized a "rain once, deploy many" model compression. EvoPress [36] proves that evolutionary search can determine suitable layerwise compression configurations and extends this method to support mixed-precision quantization and non-isotropic unstructural sparsity.

This paper presents a novel structural pruning framework, Týr-the-Pruner, for large language models. Unlike conventional methods, this framework searches for the optimal sparsity distribution within a supernet. Through enhanced supernet construction and an iterative prune-and-search technique, it achieves end-to-end global pruning optimization with strong efficiency and efficacy, setting a new benchmark for post-pruning accuracy maintenance.

# 5 Limitations

Týr-the-Pruner achieves state-of-the-art structural pruning outcomes by constructing reliable supernets and employing an iterative prune-and-search process. We have significantly reduced the search space and the number of generations explored. However, the search time cost remains non-negligible. Fair time costs in model compression are often considered acceptable, as the goal is to achieve a sufficiently optimized pruned model. However, we will continue to optimize it in future work.

# 6 Conclusion

This paper introduces Týr-the-Pruner, an end-to-end global structural pruning framework for large language models. By constructing a supernet through local pruning across various sparsity ratios and using evolutionary search to identify the optimal subnet, our framework achieves the optimal sparsity distribution under a target overall sparsity ratio. We propose an effective local pruning and an expectation error accumulation approach to enhance supernet construction. Additionally, an iterative prune-and-search strategy with coarse-to-fine sparsity granularity ensures rapid convergence. Extensive experiments show that Týr-the-Pruner outperforms state-of-the-art methods, achieving 50% parameter pruning while retaining 97% accuracy on Llama-3.1-70B.

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

# A Appendix

## A.1 Theoretical Foundations of Local Pruning

**Redundant channel identification**. We consider first- and second-order terms to minimize Equation (3). For the first-order term, we identify the to-prune channel $p$ by $\text{argmin}_{\mathbf{W}_{p,:}} \left( \left| \mathbf{G}_{p,:} \mathbf{W}_{p,:}^\top \right| \right)$, which identifies the weights with the minimal contribution in the gradient direction [23]. For the second-order term, we employ the Optimal Brain Surgeon (OBS) method [12], which optimizes $\text{argmin}_{\mathbf{W}_{p,:}} \left( \frac{\|\mathbf{W}_{p,:}\|_2^2}{2[\mathbf{H}^{-1}]_{p,p}} \right)$ by considering the inverse of the diagonal elements of the Hessian matrix. This method measures each channel's contribution to the curvature of the loss function.

The identification metric for redundant channels is derived from a manual design that takes into account both first- and second-order optimization information, distinguishing it from previous work. Table 4 demonstrates the validity of our metric by ablation.

**Weight adjustment**. We minimize Equation (3) by applying the Lagrange multiplier method to impose constraints on the $p$-th channel should be pruned ($\delta\mathbf{W}_{p,:} = \mathbf{W}_{p,:}$):

$$\mathcal{L}(\delta\mathbf{W}, \boldsymbol{\lambda}) = \mathbf{G}^\top \delta\mathbf{W} + \frac{1}{2}\delta\mathbf{W}^\top \mathbf{H}\delta\mathbf{W} + \boldsymbol{\lambda}^\top \left(\delta\mathbf{W}_{p,:} - \mathbf{W}_{p,:}\right). \tag{7}$$

Under the constraints, the resulting loss function $\mathcal{L}(\delta\mathbf{W}, \boldsymbol{\lambda})$ will be differentiated with respect to $\delta\mathbf{W}$ and $\boldsymbol{\lambda}$ to find the minimum value:

$$\begin{cases} \dfrac{\partial \mathcal{L}(\delta\mathbf{W}, \boldsymbol{\lambda})}{\partial \delta\mathbf{W}} = \mathbf{G} + \mathbf{H}\delta\mathbf{W} + \mathbf{E}_p\boldsymbol{\lambda}^\top = \mathbf{0}, \\[2mm] \dfrac{\partial \mathcal{L}(\delta\mathbf{W}, \boldsymbol{\lambda})}{\partial \boldsymbol{\lambda}} = \delta\mathbf{W}_{p,:} - \mathbf{W}_{p,:} = \mathbf{0}. \end{cases} \tag{8}$$

For the $\mathbf{G} + \mathbf{H}\delta\mathbf{W} + \mathbf{E}_p\boldsymbol{\lambda}^\top$ term, we use $p$ and $\sim p$ to denote channels to prune and channels to remain. Corresponding variables can be expanded in this way:

$$\begin{bmatrix} \mathbf{G}_{p,:} \\ \mathbf{G}_{\sim p,:} \end{bmatrix} + \begin{bmatrix} \mathbf{H}_{p,p} & \mathbf{H}_{p,\sim p} = 0 \\ \mathbf{H}_{\sim p,p} = 0 & \mathbf{H}_{\sim p,\sim p} \end{bmatrix} \begin{bmatrix} \delta\mathbf{W}_{p,:} \\ \delta\mathbf{W}_{\sim p,:} \end{bmatrix} + \begin{bmatrix} \boldsymbol{\lambda}^\top \\ \mathbf{0} \end{bmatrix} = \mathbf{0}, \tag{9}$$

where the elements of the Hessian matrix corresponding to the pruned positions $p$ can be set to zero (when a channel of the weights is pruned, the same position of the Hessian/invHessian matrix are pruned correspondingly [8]). Overall, the solution is $\delta\mathbf{W}_{\sim p,:} = -\mathbf{H}_{\sim p,\sim p}^{-1}\mathbf{G}_{\sim p,:}$.

**Fast update of inverse Hessian matrix**. When the $p$-th channel is pruned, the inverse Hessian matrix $\mathbf{H}^{-1}$ must be updated to account for the removal of the corresponding channel $p$ in $\mathbf{W}$. This update can be efficiently derived by leveraging the properties of partitioned matrices and applying the Sherman-Morrison-Woodbury formula. The main idea is that the pruning of the $p$-th channel results in a rank-1 update to $\mathbf{H}^{-1}$, which is mathematically represented as:

$$\mathbf{H}^{-1} \leftarrow \mathbf{H}^{-1} - \frac{1}{[\mathbf{H}^{-1}]_{pp}} \mathbf{H}_{:,p}^{-1} \mathbf{H}_{p,:}^{-1}. \tag{10}$$

By updating the inverse Hessian with a rank-1 adjustment, the influence of the $p$-th channel is properly removed through the outer product of the corresponding column and row vectors, using the reciprocal of the $p$-th diagonal element. The updated $\mathbf{H}^{-1}$ ensures consistency for the remaining channels, enabling efficient and scalable pruning operations. This method has a time complexity of $O(d_{\text{in}}^2)$, avoiding full recomputation of the inverse and ensuring computational efficiency.

## A.2 Algorithms for Týr-the-Pruner

**Algorithm 1** Function local_pruning

1: **Inputs:** to-prune weight $\mathbf{W}$,
         input activations $\mathbf{X}$,
         sparsity $S$,
         pruning granularity (pruning times) $K$
2: $\mathbf{Mask} \leftarrow \text{ones\_like}(\mathbf{W})$
3: $\mathbf{H} \leftarrow \mathbf{X}^{\top}\mathbf{X}$
4: $\mathbf{G} \leftarrow \mathbf{H}\mathbf{W}$
5: **for** $k \leftarrow 1$ **to** $K$ **do**
6:     $p \leftarrow \operatorname{argmin}_p \left( \left| \mathbf{G}_{p,:}\mathbf{W}_{p,:}^{\top} \right| + \frac{\|\mathbf{W}_{p,:}\|_2^2}{2\left[\mathbf{H}^{-1}\right]_{p,p}} \right)$
7:     $\mathbf{Mask}_p \leftarrow 0$
8:     $\mathbf{W}_{\sim p,:} \leftarrow \mathbf{W}_{\sim p,:} + \mathbf{H}_{\sim p,\sim p}^{-1}\mathbf{G}_{\sim p,:}$
9:     $\mathbf{H}^{-1} \leftarrow \mathbf{H}^{-1} - \frac{1}{\left[\mathbf{H}^{-1}\right]_{p,p}}\mathbf{H}_{:,p}^{-1}\mathbf{H}_{p,:}^{-1}$
10: **end for**
11: **Return** $\mathbf{Mask} \odot \mathbf{W}$

**Algorithm 2** Function prune_to_supernet

1: **Inputs:** LLM weights $\{\mathbf{W}_1, \mathbf{W}_2, ..., \mathbf{W}_L\}$,
         sparsity ratios
         $\{S_{1,1}, ..., S_{1,E}, ..., S_{L,E}\}$,
         input activations for first weight $\mathbf{X}$,
         pruning granularity (pruning times) $K$
2: **for** $\ell \leftarrow 1$ **to** $L$ **do**
3:     $\mathbf{X}\_\text{list} \leftarrow []$
4:     **for** $e \leftarrow 1$ **to** $E$ **do**
5:        $\widehat{\mathbf{W}}_{\ell,e} \leftarrow \text{local\_pruning}(\mathbf{W}_\ell, \mathbf{X}, S_{\ell,e}, K)$
6:        $\text{store}(\widehat{\mathbf{W}}_{\ell,e})$
7:        $\mathbf{X}\_\text{list. append}(\mathbf{X} \cdot \widehat{\mathbf{W}}_{\ell,e})$
8:     **end for**
9:     $\mathbf{X} \leftarrow \sum_{e=1}^{E} \frac{1-S_{\ell,e}}{\sum_{e=1}^{E}(1-S_{\ell,e})}\mathbf{X}\_\text{list}[e]$
10: **end for**
11: **Return** $\{\widehat{\mathbf{W}}_{\ell,e}\}_{\ell=1,e=1}^{L,E}$

---

**Algorithm 3** Function evolutionary_search

1: **Inputs:** sparse structures $\mathbb{W} = \{\mathbf{W}_{1,1}, ..., \mathbf{W}_{1,E}, ..., \mathbf{W}_{L,E}\}$,
         sparsity ratios $\{S_\ell\}_{\ell=1}^{L}$, sparsity interval $S^g$
2: **procedure** makeCandidates($numCanidates, \mathbb{W}, \{S_\ell\}_{\ell=1}^{L}, S^g$)
3:     $Candidates \leftarrow []$
4:     **for** $i \leftarrow 1$ **to** $numCanidates$ **do**
5:        $Candidates. \text{append}(\text{randSparsityShift}(\mathbb{W}, \{S_\ell\}_{\ell=1}^{L}, S^g, \text{randChoice}(L), \text{randChoice}(L)))$
6:     **end for**
7: **end procedure: return** $Candidates$
8: $\{\widehat{S}_\ell\}_{\ell=1}^{L} \leftarrow \{S_\ell\}_{\ell=1}^{L}$
9: **for** $g \leftarrow 1$ **to** $numGenerations$ **do**
10:     $Offsprings \leftarrow \text{makeCandidates}(numCanidates, \mathbb{W}, \{\widehat{S}_\ell\}_{\ell=1}^{L}, S^g)$
11:     $\{\widehat{S}_\ell\}_{\ell=1}^{L} \leftarrow \text{checkSparsity}(\text{argminSearchMetric}(Offsprings))$
12: **Return** $\{\widehat{S}_\ell\}_{\ell=1}^{L}$

---

**Algorithm 4** Function Týr-the-Pruner

1: **Inputs:** LLM weights $\{\mathbf{W}_1, \mathbf{W}_2, ..., \mathbf{W}_L\}$, input activations for first weight $\mathbf{X}$,
         pruning granularity (pruning times) $K$, overall sparsity $S$, sparsity interval $S^g$,
         num sparse structures $E$, iterations $T$
2: **procedure** generateSparsities($L, E, \{S_\ell\}_{\ell=1}^{L}, S^g$)
3:     $Sparsities = \{\}$
4:     **for** $\ell \leftarrow 0$ **to** $\text{range}(L)$ **do**
5:        **for** $e \leftarrow 0$ **to** $\text{range}(E)$ **do**
6:           $Sparsities. \text{append}(S_\ell - ((e-1) \times 0.5) \times S^g + i \times S^g)$
7:        **end for**
8:     **end for**
9: **end procedure: return** $Sparsities$
10: $\{\widehat{S}_\ell\}_{\ell=1}^{L} \leftarrow \{S\}_L$
11: **for** $t \leftarrow 1$ **to** $T$ **do**
12:     $Sparsities \leftarrow \text{generateSparsities}(L, E, \{\widehat{S}_\ell\}_{\ell=1}^{L}, S^g)$
13:     $\{\widehat{\mathbf{W}}_{\ell,e}\}_{\ell=1,e=1}^{L,E} \leftarrow \text{prune\_to\_supernet}(\{\mathbf{W}_\ell\}_{\ell=1}^{L}, Sparsities, \mathbf{X}, K)$
14:     $\{\widehat{S}_\ell\}_{\ell=1}^{L} \leftarrow \text{evolutionary\_search}(\{\widehat{\mathbf{W}}_{\ell,e}\}_{\ell=1,e=1}^{L,E}, \{\widehat{S}_\ell\}_{\ell=1}^{L}, S^g)$
15:     $S^g \leftarrow S^g \times 0.5$
16: **end for**
17: **Return** $\text{compress}(\{\widehat{\mathbf{W}}_{\ell,e}\}_{\ell=1,e=1}^{L,E}, \{\widehat{S}_\ell\}_{\ell=1}^{L})$

## A.3 Further Comparisons

To further demonstrate the effectiveness of our proposed method, Týr-the-Pruner, we conducted a more comprehensive comparison. The competitors include the pure subnet search framework SearchLLM [35], the probe-based dynamic pruning approach ProbePruning [19], the sparsity distribution optimizer Adapt-Pruner [44], the coarse-and-fine combined approach CFSP [46], the calibration-free approach PruneNet [19], the structure-independent approach DISP-LLM [11], the cluster-based evolutionary pruning approach EvoP [48], and the search-only approach DarwinLLM [38]. The experimental results, with competitor performance taken from their respective papers, are presented in Table 9.

It is evident that Týr-the-Pruner significantly outperforms other structured pruning methods,

Table 9: **Further comparisons**. Perplexity on Wikitext2 (lower is better) and 0-shot accuracy (%, higher is better, DarwinLLM reported the 25-shot Arc-C benchmark) serve as the comparison metrics. Optimal results are **bolded**.

| Model | Sparsity | Method | Wikitext2 ↓ | BoolQ ↑ | WinoGrande ↑ | ARC-E ↑ | ARC-C ↑ |
|---|---|---|---|---|---|---|---|
| Llama-7B | 0% | N/A | 5.68 | 71.38 | 67.01 | 67.45 | 41.38 |
| | 20% | SearchLLM | **6.89** | 70.98 | 74.92 | 64.23 | 36.52 |
| | 25% | Týr-the-Pruner | 7.36 | **75.81** | **75.68** | **66.36** | **42.06** |
| Llama-2-7B | 0% | N/A | 5.12 | 77.68 | 69.06 | 76.30 | 43.43 |
| | 30% | PruneNet | - | - | 61.09 | 53.20 | 33.53 |
| | | DISP-LLM | 6.85 | - | 62.27 | 59.81 | 33.19 |
| | 37.5% | Týr-the-Pruner | 10.29 | **68.87** | **66.93** | **71.13** | **38.31** |
| | 40% | ProbePruning | **8.01** | 64.70 | 58.10 | 62.50 | 37.70 |
| | 50% | DISP-LLM | **9.84** | - | 58.41 | 43.06 | 25.85 |
| | | DarwinLM | - | 62.70 | 55.80 | 63.30 | 38.10 |
| | | Týr-the-Pruner | 16.17 | 65.54 | 62.12 | 66.12 | 33.62 |
| Llama-2-13B | 0% | N/A | 4.57 | 80.61 | 72.22 | 79.46 | 48.46 |
| | 20% | EvoP | 6.33 | - | 68.00 | 73.00 | 40.00 |
| | 25% | Týr-the-Pruner | **5.79** | **81.35** | **72.06** | **77.74** | **44.97** |
| | 30% | DISP-LLM | **5.77** | - | 66.85 | 63.80 | 39.42 |
| | 37.5% | Týr-the-Pruner | 7.17 | 80.76 | **72.06** | **76.35** | **43.26** |
| | 50% | CFSP | - | - | 64.17 | 62.33 | 38.05 |
| | | DISP-LLM | **7.11** | - | 59.27 | 52.57 | 33.28 |
| | | Týr-the-Pruner | 9.59 | 74.46 | 70.09 | 72.18 | 39.85 |
| Llama-3.1-8B | 0% | N/A | 5.84 | 82.17 | 73.56 | 81.31 | 51.54 |
| | 40% | Adapt-Pruner | 33.75 | - | 56.75 | 45.16 | 25.97 |
| | 50% | DarwinLM | - | 62.20 | 57.30 | 59.60 | 34.20 |
| | | Týr-the-Pruner | **30.89** | **66.64** | **61.80** | **65.86** | 31.83 |
| Llama-3-8B | 0% | N/A | 5.76 | 81.10 | 73.01 | 80.05 | 50.43 |
| | 40% MLP-only | ProbePruning | 14.90 | 70.30 | 67.20 | 57.40 | 39.00 |
| | 25% | Týr-the-Pruner | **13.14** | **76.02** | **71.11** | **75.63** | **42.15** |

achieving better performance even at higher sparsities compared to other methods at lower sparsities. In particular, Týr-the-Pruner surpasses the search-based methods SearchLLM, EvoP, and DarwinLLM, demonstrating the effectiveness of our effective local pruning approach, expected error accumulation, and iterative prune-and-search strategy.

## A.4 Efficiency Analysis on Non-isotropic Structural Pruning

Large language models (LLMs) with non-isotropic pruning may be considered to exhibit inferior inference efficiency compared to those with isotropic sparsity across layers. To explore, we provide a comparative analysis of inference efficiency for Llama-3.1-8B and Mistral-Nemo, both pre- and post-50% structural pruning. The evaluation was conducted on an AMD Instinct™ MI250 Accelerator using Pytorch (HipBlas), covering both prefilling and decoding tasks across a range of sentence lengths, as illustrated in Figure 4.

The variance (Var) quantifies the degree of variation in sparsity under non-isotropic pruning conditions; a larger variance indicates more fluctuation in sparsity across layers. As shown in Figure 4, the 50% structural pruned LLMs achieve up to 1.3x or greater speedup in both prefilling and decoding tasks compared to their dense counterparts across most sentence lengths. Variations in layer sparsity do not have a significant impact on efficiency. A slight efficiency decrease is only observed when the variance reaches 1. In this case, the reduction in efficiency is likely due to the frequent high sparsity, which leads to more memory-bottlenecked "thin" matrix multiplications in the computational graph.

Figure 5(a) and Figure 5(b) compare the sparsity distributions of the MHA and FFN layers in Llama-3.1-8B after 50% pruning with Týr-the-Pruner and the search-only methods, respectively. The sparsity distribution obtained by Týr-the-Pruner resembles that of the search-only strategy, yet Týr-the-Pruner performs better. Its search process is more refined, incorporating multiple rounds of expectation error accumulation, ultimately leading to a superior sparsity distribution and higher performance in the pruned model.

Figure 5(c) and Figure 5(d) compare the sparsity distributions of the MHA and FFN layers in the 50% post-pruned Llama-3.1-8B across different iterations of Týr-the-Pruner. Týr-the-Pruner identifies a

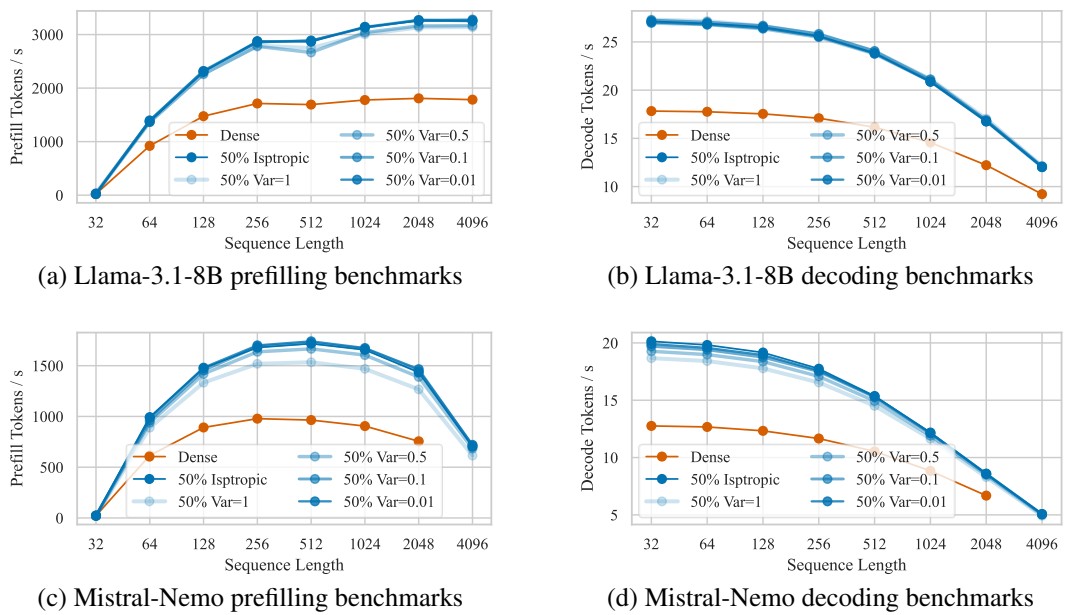

(a) Llama-3.1-8B prefilling benchmarks

(b) Llama-3.1-8B decoding benchmarks

(c) Mistral-Nemo prefilling benchmarks

(d) Mistral-Nemo decoding benchmarks

Figure 4: Pre- and post-pruning large language model inference benchmarks.

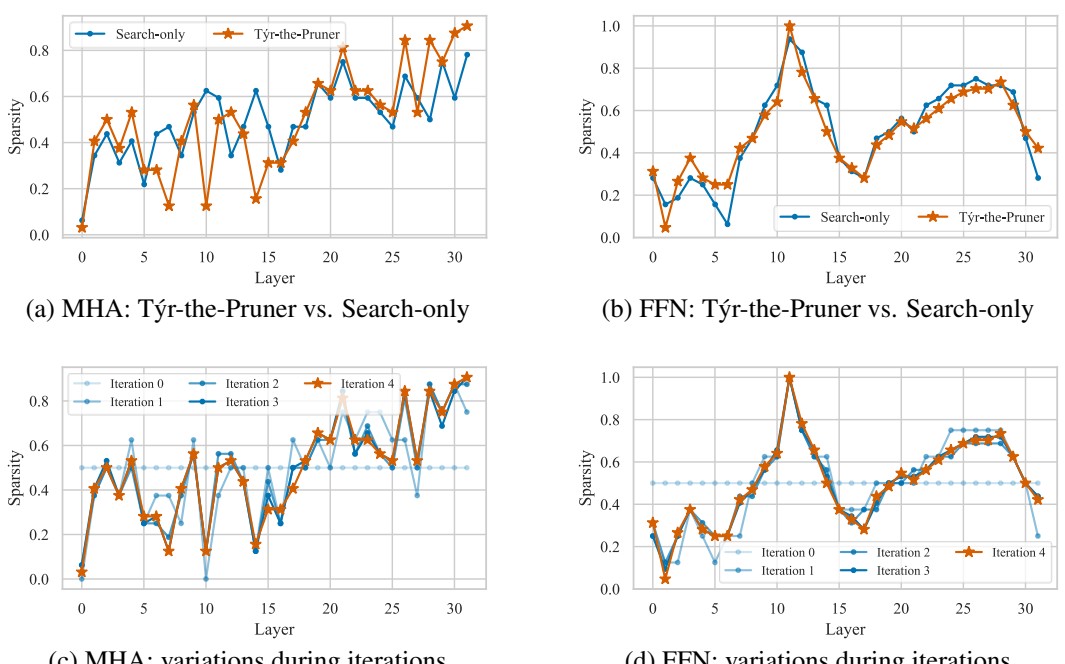

(a) MHA: Týr-the-Pruner vs. Search-only

(b) FFN: Týr-the-Pruner vs. Search-only

(c) MHA: variations during iterations

(d) FFN: variations during iterations

Figure 5: Sparsity distribution of Týr-the-Pruner and the search-only strategy on Llama-3.1-8B.

relatively ideal and coarse-grained sparsity distribution in the first search (with a sparsity interval of 12.5%). In the subsequent iterations (2nd, 3rd, and 4th), with sparsity intervals of 6.25%, 3.125%, and 1.5625%, respectively, the sparsity distribution is progressively refined and optimized, ultimately converging to an optimal solution.

## A.5 Sparsity Distribution of Different Pruning Methods

Different pruning methods vary in the distribution of sparsity. Figure 6(a) and Figure 6(b) show the sparsity distributions of MHA and FFN of Llama-3.1-8B after 50% pruning by a series of LLM structural pruning methods, respectively.

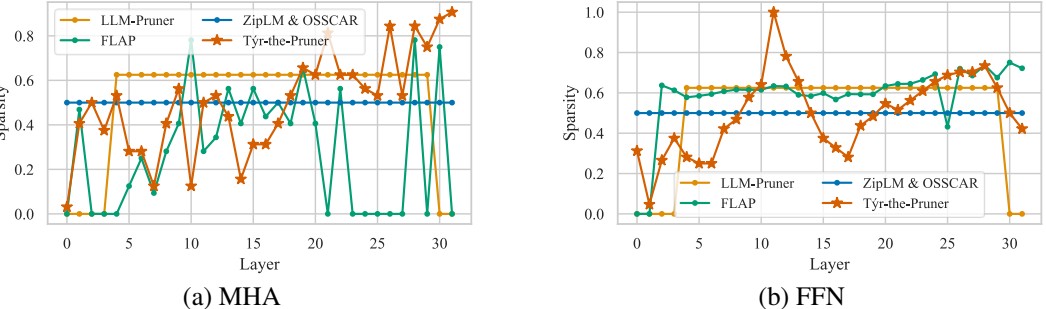

(a) MHA          (b) FFN

Figure 6: Sparsity distributions with different structural pruning methods.

ZipLM and OSSCAR maintain isotropic sparsity distribution. LLM-Pruner incorporates prior knowledge, recognizing that the shallow and deep layers of LLMs are more pruning-sensitive and thus preserve them while only isotropically pruning the intermediate layers. These three methods fail to account for the unique characteristics of different LLMs, leading to clear suboptimal sparsity assignments. Conversely, FLAP combines local activations and weights to assess the global sparsity distribution, resulting in non-isotropic pruning. While this method seeks a balance between local and global sparsity, it does not fully address the gap between them, making it challenging to achieve an optimal sparsity distribution.

Týr-the-Pruner's sparsity distribution clearly differs from that of other methods. It directly searches for the optimal sparsity distribution at the global level without the local and global gaps. The resulting sparsity distribution does not adhere to prior assumptions: for instance, the 2-nd FFN layer is largely retained, while the 12-th FFN layer is entirely pruned, and there is no discernible pattern in the sparsity ratio as layers become deeper or shallower. This demonstrates that model optimization should fully account for the model's unique characteristics.

## A.6 Prune and Tune

Table 10: **Results of pruning and finetuning**. Perplexity on Wikitext2 (lower is better) and accuracy (%, higher is better) serve as the comparison metrics. MMLU employed a 5-shot benchmark, while other tasks used 0-shot benchmarks. An asterisk (*) indicates a fine-tuned result. Optimal and suboptimal results are **bolded** and underlined, respectively.

| Sparsity | Method | Wikitext2 ↓ | Avg. Acc. ↑ | Arc-C ↑ | Arc-E ↑ | BoolQ ↑ | HellaSwag ↑ | OBQA ↑ | RTE ↑ | WinoGrande ↑ | MMLU ↑ |
|---|---|---|---|---|---|---|---|---|---|---|---|
| 0% | N/A | 5.84 | 64.77 | 51.54 | 81.31 | 82.17 | 60.04 | 33.20 | 71.12 | 73.56 | 65.20 |
| 37.5% | LLM-Pruner | 70.93 | 32.87 | 19.68 | 32.07 | 40.03 | 27.55 | 13.20 | 52.71 | 50.83 | 26.88 |
| | LLM-Pruner* | 17.97 | 47.11 | 32.51 | 64.18 | 63.09 | 47.79 | 26.00 | 55.23 | 58.56 | 29.53 |
| | NutePrune* | 14.31 | 51.33 | 36.77 | 62.92 | 68.78 | 49.45 | 29.20 | 58.12 | 66.93 | 38.49 |
| | FLAP | 21.54 | 43.07 | 23.98 | 52.15 | 64.62 | 36.50 | 23.40 | 58.17 | 58.88 | 30.17 |
| | FLAP* | 18.47 | 48.43 | 34.39 | 62.33 | 65.87 | 44.77 | 28.80 | 56.68 | 58.88 | 35.74 |
| | Týr-the-Pruner | 18.09 | 53.46 | 39.68 | 73.53 | 70.55 | 47.12 | 30.00 | 58.84 | 66.54 | 41.43 |
| | Týr-the-Pruner* | **12.65** | **58.22** | **46.93** | **77.02** | **75.75** | **54.99** | **32.60** | **59.57** | **68.82** | **50.06** |
| 50% | LLM-Pruner | 288.32 | 31.58 | 19.62 | 28.70 | 37.83 | 26.36 | 13.40 | 52.35 | 49.64 | 24.70 |
| | LLM-Pruner* | 27.34 | 40.39 | 25.68 | 53.49 | 45.84 | 39.88 | 22.00 | 53.79 | 55.25 | 27.22 |
| | NutePrune* | 23.55 | 42.71 | 30.01 | 53.84 | 55.61 | 38.02 | 24.20 | 55.23 | 58.25 | 26.50 |
| | FLAP | 134.28 | 36.59 | 20.99 | 43.18 | 52.29 | 29.43 | 16.80 | 52.71 | 54.14 | 23.18 |
| | FLAP* | 51.29 | 43.01 | 29.18 | 53.32 | 60.83 | 37.15 | 22.00 | 56.68 | 57.14 | 27.77 |
| | Týr-the-Pruner | 30.89 | 47.79 | 31.83 | 65.36 | 66.64 | 39.99 | 24.80 | 58.12 | 61.80 | 33.76 |
| | Týr-the-Pruner* | **19.68** | **51.83** | **38.65** | **70.92** | **67.25** | **48.62** | **31.60** | **60.29** | **62.12** | **35.22** |

Structured pruning is often followed by parameter-efficient fine-tuning to restore model performance [23, 40, 20]. To evaluate the benefits of our proposed method, Týr-the-Pruner, in the context of

post-pruning fine-tuning, we conducted fine-tuning experiments on the Llama-3.1-8B model. The fine-tuning employed the parameter-efficient method LoRA (rank=16) [14], and the post-pruned LLM was fine-tuned for three epochs on the Alpaca-GPT4 dataset [32].

Table 10 presents the experimental results. Týr-the-Pruner demonstrates superior performance at both 37.5% and 50% sparsity, with many of the results without fine-tuning already outperforming those fine-tuned by other methods. Furthermore, Týr-the-Pruner surpasses the state-of-the-art training-aware pruning method, NutePrune [20].

## A.7    Statistical Significance Analysis

To verify the robustness of the proposed method, Týr-the-Pruner, we adjust the random seeds (the change of random seeds triggers the change of calibration samples and the change in random sparsity shift) for multiple (number of tests: n=5) experiments and observe the error bar ($\pm$ standard deviation), as shown in Table 11.

Table 11: Statistical significance analysis for Týr-the-Pruner.

| Model | Sparsity | Wikitext2 $\downarrow$ | BoolQ $\uparrow$ | ARC-E $\uparrow$ | ARC-C $\uparrow$ |
|---|---|---|---|---|---|
| Llama-2-7B | 25% | $7.51 \pm 0.07$ | $69.45 \pm 0.04$ | $75.13 \pm 0.10$ | $42.58 \pm 0.09$ |
| Llama-2-13B | 25% | $5.79 \pm 0.00$ | $81.35 \pm 0.06$ | $77.74 \pm 0.03$ | $44.97 \pm 0.05$ |
| Llama-3.1-8B | 25% | $10.38 \pm 0.11$ | $76.36 \pm 0.12$ | $77.23 \pm 0.09$ | $45.48 \pm 0.06$ |
| | 50% | $30.89 \pm 0.21$ | $66.64 \pm 0.26$ | $65.86 \pm 0.33$ | $31.83 \pm 0.16$ |

From the global observation of experimental results, the proposed method performs relatively consistently in multiple randomized trials, with standard deviations within acceptable limits ($< 0.21$ for Wikitext2 perplexity and $< 0.33$ for downstream performance). From the local observation of experimental results, it can be seen that pruning yields a more stable performance for larger models or under lower sparsity ratios.

## A.8 Detailed Downstream Task Results

### Table 12: 0-shot acc (%) on ARC-Challenge.

| Sparsity | Method | LLaMA-2 | | LLaMA-3.x | | | Mistral | |
|---|---|---|---|---|---|---|---|---|
| | | 7B | 13B | 2-3B | 0-8B | 1-8B | 7B-v0.3 | Nemo |
| 0% | N/A | 43.43 | 48.46 | 42.32 | 50.43 | 51.54 | 48.81 | 55.72 |
| 12.5% | ShortGPT | 36.18 | 43.86 | 37.80 | 43.94 | 44.20 | 43.43 | 46.16 |
| | LaCO+ | 37.97 | 43.60 | 37.63 | 43.69 | 44.20 | 42.15 | 45.14 |
| | SliceGPT | 41.81 | 46.25 | 35.15 | 41.64 | 42.15 | 42.49 | 31.66 |
| | Wanda-sp | **43.34** | 45.05 | 24.06 | 19.28 | 34.04 | 46.16 | 48.89 |
| | LLM-Pruner | 38.40 | 44.45 | 31.83 | 38.57 | 37.97 | 40.02 | 43.52 |
| | ZipLM | 41.55 | 49.15 | 38.23 | 40.19 | 42.49 | 47.27 | 52.30 |
| | OSSCAR | 42.41 | **49.23** | 38.14 | 40.70 | 40.78 | 46.59 | 30.80 |
| | FLAP | 40.02 | 42.15 | 33.45 | 41.30 | 41.47 | 43.86 | 45.48 |
| | Týr-the-Pruner | 42.06 | 48.05 | **38.82** | **47.44** | **49.15** | **48.55** | **54.35** |
| 25% | ShortGPT | 32.34 | 37.88 | 30.97 | 26.88 | 27.39 | 33.96 | 38.99 |
| | LaCO+ | 30.89 | 38.65 | 32.34 | 36.26 | 36.77 | 33.87 | 40.10 |
| | SliceGPT | 37.88 | 41.47 | 28.33 | 35.32 | 37.46 | 38.99 | 24.49 |
| | Wanda-sp | 38.14 | 20.82 | 18.09 | 16.89 | 19.20 | 37.37 | 23.21 |
| | LLM-Pruner | 28.24 | 37.63 | 22.35 | 26.11 | 24.57 | 31.66 | 31.40 |
| | ZipLM | 39.51 | **46.93** | 29.52 | 18.43 | 20.31 | 43.69 | 27.39 |
| | OSSCAR | 40.53 | 45.65 | 18.17 | 27.47 | 23.46 | 42.66 | 27.22 |
| | FLAP | 31.91 | 40.78 | 26.54 | 31.91 | 33.45 | 36.52 | 40.70 |
| | Týr-the-Pruner | **42.58** | 44.97 | **35.41** | **42.15** | **45.48** | **44.88** | **48.38** |
| 37.5% | ShortGPT | 28.58 | 31.14 | 25.85 | 26.88 | 27.56 | 29.10 | 27.90 |
| | LaCO+ | 28.24 | 32.08 | 25.43 | 27.30 | 27.05 | 28.24 | 31.57 |
| | SliceGPT | 32.00 | 36.60 | 23.29 | 27.65 | 27.39 | 28.67 | 19.45 |
| | Wanda-sp | 27.99 | 21.76 | 20.39 | 20.90 | 19.97 | 20.65 | 20.31 |
| | LLM-Pruner | 17.58 | 24.40 | 17.49 | 16.89 | 16.98 | 20.90 | 19.28 |
| | ZipLM | 33.53 | 32.08 | 20.56 | 19.80 | 21.16 | 38.31 | 37.03 |
| | OSSCAR | 35.49 | 33.96 | 18.77 | 26.54 | 23.98 | 36.77 | 37.63 |
| | FLAP | 29.18 | 35.75 | 24.40 | 25.17 | 23.98 | 29.69 | 32.51 |
| | Týr-the-Pruner | **38.31** | **43.26** | **30.97** | **38.99** | **39.68** | **38.31** | **42.41** |
| 50% | ShortGPT | 23.46 | 28.16 | 21.84 | 23.29 | 23.56 | 26.19 | 22.23 |
| | LaCO+ | 23.55 | 27.05 | 21.25 | 24.74 | 22.87 | 24.49 | 21.67 |
| | SliceGPT | 24.91 | 30.63 | 18.86 | 20.99 | 21.50 | 19.45 | 18.52 |
| | Wanda-sp | 17.58 | 19.54 | 20.73 | 18.60 | 19.54 | 18.00 | 18.00 |
| | LLM-Pruner | 18.60 | 19.54 | 19.11 | 17.32 | 19.62 | 18.52 | 21.59 |
| | ZipLM | 20.14 | 27.99 | 19.20 | 17.15 | 20.48 | 23.72 | 21.16 |
| | OSSCAR | 23.81 | 25.34 | 20.56 | 17.58 | 19.97 | 27.13 | 20.82 |
| | FLAP | 29.10 | 27.47 | 22.78 | 21.76 | 20.99 | 25.34 | 28.24 |
| | Týr-the-Pruner | **33.62** | **39.85** | **25.51** | **32.34** | **31.83** | **32.94** | **32.59** |

### Table 13: 0-shot acc (%) on ARC-Easy.

| Sparsity | Method | LLaMA-2 | | LLaMA-3.x | | | Mistral | |
|---|---|---|---|---|---|---|---|---|
| | | 7B | 13B | 2-3B | 0-8B | 1-8B | 7B-v0.3 | Nemo |
| 0% | N/A | 76.30 | 79.46 | 74.49 | 80.05 | 81.31 | 79.67 | 83.00 |
| 12.5% | ShortGPT | 65.87 | 75.55 | 68.10 | 71.17 | 72.18 | 71.51 | 75.84 |
| | LaCO+ | 68.39 | 75.04 | 64.69 | 73.44 | 75.67 | 71.25 | 75.67 |
| | SliceGPT | 73.40 | 77.78 | 67.68 | 74.71 | 75.51 | 76.85 | 53.70 |
| | Wanda-sp | 74.62 | 76.43 | 48.95 | 28.70 | 64.44 | 77.61 | 79.29 |
| | LLM-Pruner | 72.05 | 77.10 | 62.92 | 70.83 | 72.47 | 73.32 | 75.46 |
| | ZipLM | 75.72 | **79.80** | 71.51 | 73.74 | 75.34 | 78.62 | 79.63 |
| | OSSCAR | **76.01** | 79.59 | 71.55 | 74.37 | 76.05 | 78.28 | 52.90 |
| | FLAP | 71.38 | 72.69 | 64.44 | 73.36 | 74.16 | 75.76 | 75.88 |
| | Týr-the-Pruner | 75.84 | 79.62 | **72.94** | **79.08** | **79.80** | **79.84** | **81.61** |
| 25% | ShortGPT | 52.74 | 61.24 | 49.58 | 38.85 | 43.18 | 52.57 | 63.30 |
| | LaCO+ | 53.03 | 64.73 | 49.41 | 53.41 | 55.47 | 52.23 | 62.50 |
| | SliceGPT | 71.80 | 74.92 | 58.67 | 67.80 | 68.60 | 71.46 | 47.56 |
| | Wanda-sp | 70.41 | 33.59 | 37.46 | 42.26 | 28.41 | 70.83 | 53.66 |
| | LLM-Pruner | 59.97 | 70.20 | 50.72 | 59.43 | 57.79 | 65.87 | 64.02 |
| | ZipLM | 74.66 | **78.45** | 61.32 | 27.86 | 26.47 | 75.88 | 50.04 |
| | OSSCAR | 74.45 | 77.57 | 27.95 | 53.70 | 40.03 | 75.59 | 51.60 |
| | FLAP | 64.23 | 69.23 | 53.96 | 60.31 | 65.95 | 67.22 | 68.69 |
| | Týr-the-Pruner | **75.13** | 77.74 | **69.40** | **75.63** | **77.23** | **77.23** | **80.13** |
| 37.5% | ShortGPT | 41.58 | 48.95 | 40.07 | 37.50 | 39.94 | 33.88 | 42.72 |
| | LaCO+ | 36.11 | 48.57 | 40.49 | 39.27 | 40.45 | 35.40 | 46.09 |
| | SliceGPT | 62.75 | 67.47 | 46.89 | 55.72 | 57.49 | 58.46 | 40.45 |
| | Wanda-sp | 57.03 | 32.37 | 26.94 | 25.72 | 25.00 | 47.90 | 35.90 |
| | LLM-Pruner | 38.93 | 54.76 | 31.69 | 32.53 | 32.07 | 47.10 | 47.98 |
| | ZipLM | 68.48 | 61.95 | 27.99 | 27.10 | 27.02 | 70.54 | 70.79 |
| | OSSCAR | 68.90 | 62.16 | 28.03 | 54.21 | 47.10 | 71.04 | 70.08 |
| | FLAP | 53.45 | 58.16 | 46.55 | 46.63 | 52.15 | 56.65 | 61.70 |
| | Týr-the-Pruner | **71.13** | **76.35** | **64.52** | **72.56** | **73.53** | **71.38** | **75.51** |
| 50% | ShortGPT | 32.45 | 37.75 | 30.39 | 31.23 | 32.83 | 32.87 | 36.28 |
| | LaCO+ | 30.01 | 37.71 | 29.00 | 28.58 | 28.54 | 30.18 | 28.96 |
| | SliceGPT | 48.40 | 54.84 | 36.20 | 41.33 | 41.62 | 43.56 | 35.23 |
| | Wanda-sp | 27.95 | 35.86 | 26.89 | 30.98 | 30.47 | 32.79 | 35.94 |
| | LLM-Pruner | 28.11 | 33.54 | 24.49 | 28.24 | 28.70 | 28.70 | 28.16 |
| | ZipLM | 29.38 | 54.00 | 27.57 | 25.57 | 28.28 | 50.84 | 49.96 |
| | OSSCAR | 50.72 | 45.92 | 27.15 | 28.07 | 26.05 | 59.22 | 41.92 |
| | FLAP | 47.01 | 43.18 | 27.23 | 42.30 | 43.18 | 52.61 | 52.57 |
| | Týr-the-Pruner | **66.12** | **72.18** | **56.23** | **65.36** | **65.36** | **66.37** | **66.04** |

### Table 14: 0-shot acc (%) on BoolQ.

| Sparsity | Method | LLaMA-2 | | LLaMA-3.x | | | Mistral | |
|---|---|---|---|---|---|---|---|---|
| | | 7B | 13B | 2-3B | 0-8B | 1-8B | 7B-v0.3 | Nemo |
| 0% | N/A | 77.68 | 80.61 | 73.00 | 81.10 | 82.17 | 82.17 | 85.14 |
| 12.5% | ShortGPT | 74.77 | 75.84 | 63.30 | 73.70 | 70.70 | 77.31 | 66.21 |
| | LaCO+ | 61.13 | 68.90 | 62.72 | 72.78 | 70.06 | 77.19 | 68.62 |
| | SliceGPT | 73.12 | 80.67 | 68.99 | 75.75 | 75.57 | 81.19 | 77.71 |
| | Wanda-sp | 71.68 | 77.28 | 51.90 | 53.64 | 63.09 | 77.31 | 68.04 |
| | LLM-Pruner | **76.48** | 80.43 | 65.72 | 74.34 | 71.90 | 72.72 | 77.58 |
| | ZipLM | 69.36 | 82.84 | 65.60 | 75.63 | 77.00 | **82.26** | 71.83 |
| | OSSCAR | 69.02 | **83.00** | 68.59 | 74.80 | 79.91 | 81.53 | 73.03 |
| | FLAP | 70.98 | 76.21 | 60.06 | 73.49 | 71.87 | 77.49 | 80.24 |
| | Týr-the-Pruner | 70.67 | 82.78 | **72.32** | **80.12** | **80.24** | 82.11 | **82.94** |
| 25% | ShortGPT | 62.17 | 62.54 | 44.83 | 37.80 | 37.65 | 67.25 | 67.22 |
| | LaCO+ | 50.83 | 58.23 | **70.64** | 63.85 | 59.14 | 75.14 | 66.70 |
| | SliceGPT | 68.93 | 79.27 | 65.81 | 72.02 | 67.68 | 75.78 | 68.41 |
| | Wanda-sp | 68.96 | 62.17 | 46.02 | 48.90 | 42.17 | 62.45 | 61.93 |
| | LLM-Pruner | 62.97 | 68.35 | 61.59 | 60.89 | 57.89 | 68.78 | 64.25 |
| | ZipLM | 67.19 | 81.31 | 59.20 | 56.02 | 65.08 | 77.16 | 65.14 |
| | OSSCAR | 66.42 | 79.48 | 54.28 | 60.06 | 65.66 | 77.13 | 64.28 |
| | FLAP | 65.47 | 68.81 | 64.89 | 68.29 | 67.28 | 65.14 | 63.82 |
| | Týr-the-Pruner | 69.45 | **81.35** | 67.89 | **76.02** | **76.36** | **79.39** | **82.26** |
| 37.5% | ShortGPT | 62.17 | 37.25 | **68.87** | 56.57 | 55.66 | 45.60 | 58.99 |
| | LaCO+ | 62.11 | 62.78 | 63.30 | 48.78 | 45.38 | 63.12 | 64.62 |
| | SliceGPT | 63.00 | 71.44 | 42.08 | 50.49 | 46.85 | 65.41 | 60.06 |
| | Wanda-sp | 62.26 | 62.17 | 52.66 | 38.13 | 51.68 | 62.05 | 49.97 |
| | LLM-Pruner | 61.74 | 62.11 | 50.70 | 41.31 | 40.03 | 62.35 | 61.87 |
| | ZipLM | 64.89 | 76.79 | 49.76 | 51.56 | 61.47 | 69.91 | 62.72 |
| | OSSCAR | 64.65 | 74.25 | 49.54 | 58.01 | 62.26 | 67.37 | 62.40 |
| | FLAP | 63.46 | 65.60 | 61.93 | 62.66 | 64.62 | 62.54 | 65.50 |
| | Týr-the-Pruner | **68.87** | **80.76** | 66.33 | **70.09** | **70.55** | **70.85** | **74.65** |
| 50% | ShortGPT | 62.17 | 62.20 | 46.61 | 62.57 | 62.17 | 51.90 | 55.29 |
| | LaCO+ | 54.83 | 59.51 | 44.40 | 55.32 | 51.41 | 42.66 | 46.57 |
| | SliceGPT | 57.16 | 62.26 | 40.76 | 41.74 | 38.56 | 51.13 | 51.53 |
| | Wanda-sp | 46.91 | 62.14 | 41.59 | 54.77 | 40.37 | 48.99 | 43.06 |
| | LLM-Pruner | 38.23 | 61.31 | 38.10 | 39.54 | 37.83 | 43.24 | 43.94 |
| | ZipLM | 43.79 | 64.80 | 44.95 | 54.19 | 57.43 | 62.72 | 62.23 |
| | OSSCAR | 61.62 | 62.94 | 56.48 | 53.36 | 61.04 | 60.95 | 62.17 |
| | FLAP | 58.50 | 65.14 | 51.25 | 61.65 | 52.29 | 61.47 | 48.87 |
| | Týr-the-Pruner | **65.54** | **74.46** | **62.26** | **65.63** | **66.64** | 62.17 | 65.26 |

### Table 15: 0-shot acc (%) on HellaSwag.

| Sparsity | Method | LLaMA-2 | | LLaMA-3.x | | | Mistral | |
|---|---|---|---|---|---|---|---|---|
| | | 7B | 13B | 2-3B | 0-8B | 1-8B | 7B-v0.3 | Nemo |
| 0% | N/A | 57.14 | 60.04 | 55.20 | 60.11 | 60.04 | 60.92 | 62.90 |
| 12.5% | ShortGPT | 49.88 | 55.70 | 49.76 | 55.12 | 55.09 | 54.93 | 56.26 |
| | LaCO+ | 51.11 | 56.22 | 48.94 | 54.51 | 54.68 | 54.79 | 55.95 |
| | SliceGPT | 52.32 | 56.16 | 47.73 | 52.03 | 50.97 | 54.57 | 50.66 |
| | Wanda-sp | **56.53** | 53.96 | 34.60 | 27.29 | 40.76 | 55.97 | 52.57 |
| | LLM-Pruner | 51.06 | 57.60 | 43.62 | 49.63 | 50.02 | 51.28 | 52.34 |
| | ZipLM | 55.41 | 59.42 | 48.85 | 51.87 | 52.37 | 57.92 | 54.77 |
| | OSSCAR | 55.39 | **59.53** | 48.60 | 51.35 | 55.21 | 57.83 | 54.05 |
| | FLAP | 53.98 | 57.21 | 43.88 | 50.97 | 51.66 | 54.20 | 51.15 |
| | Týr-the-Pruner | 55.88 | 59.39 | **51.55** | **56.52** | **56.32** | **58.27** | **59.31** |
| 25% | ShortGPT | 41.94 | 47.70 | 37.31 | 28.89 | 28.37 | 42.51 | 43.61 |
| | LaCO+ | 42.16 | 49.34 | 39.36 | 43.97 | 43.92 | 42.73 | 45.33 |
| | SliceGPT | 46.16 | 49.84 | 39.29 | 43.29 | 42.10 | 44.24 | 40.37 |
| | Wanda-sp | 51.21 | 34.47 | 28.87 | 28.04 | 27.25 | 44.23 | 34.65 |
| | LLM-Pruner | 38.80 | 46.81 | 32.91 | 33.94 | 33.05 | 38.66 | 38.11 |
| | ZipLM | 51.57 | 55.93 | 33.39 | 32.32 | 30.47 | 51.34 | 43.72 |
| | OSSCAR | 51.61 | 55.16 | 26.55 | 36.45 | 36.44 | 50.69 | 43.09 |
| | FLAP | 47.73 | 51.42 | 37.10 | 42.54 | 43.16 | 45.80 | 44.20 |
| | Týr-the-Pruner | **52.86** | **57.49** | **46.62** | **53.10** | **52.87** | **58.27** | **55.04** |
| 37.5% | ShortGPT | 33.53 | 39.31 | 31.44 | 32.11 | 30.71 | 27.72 | 34.81 |
| | LaCO+ | 33.56 | 41.75 | 33.12 | 34.16 | 34.07 | 31.45 | 32.18 |
| | SliceGPT | 37.65 | 41.27 | 32.08 | 34.14 | 33.37 | 34.50 | 32.48 |
| | Wanda-sp | 35.07 | 29.49 | 26.48 | 25.66 | 26.38 | 30.63 | 26.08 |
| | LLM-Pruner | 28.17 | 33.06 | 26.68 | 27.52 | 27.55 | 29.38 | 28.36 |
| | ZipLM | 38.29 | 45.79 | 26.89 | 29.11 | 27.43 | 40.65 | 35.69 |
| | OSSCAR | 42.86 | 48.20 | 26.66 | 30.70 | 31.49 | 41.37 | 35.09 |
| | FLAP | 41.53 | 45.52 | 32.69 | 36.48 | 36.50 | 37.49 | 39.28 |
| | Týr-the-Pruner | **48.47** | **54.11** | **39.97** | **47.22** | **47.12** | **46.01** | **48.22** |
| 50% | ShortGPT | 28.61 | 32.44 | 28.01 | 27.82 | 27.87 | 26.30 | 30.83 |
| | LaCO+ | 27.72 | 31.64 | 28.31 | 27.71 | 26.01 | 28.28 | 27.51 |
| | SliceGPT | 30.91 | 32.35 | 28.22 | 28.96 | 29.07 | 29.60 | 29.02 |
| | Wanda-sp | 26.65 | 28.52 | 26.32 | 26.72 | 26.73 | 27.61 | 26.11 |
| | LLM-Pruner | 26.76 | 27.78 | 26.60 | 26.43 | 26.34 | 27.09 | 25.96 |
| | ZipLM | 26.53 | 35.84 | 26.46 | 27.52 | 26.42 | 32.17 | 30.51 |
| | OSSCAR | 32.21 | 32.16 | 26.58 | 27.81 | 26.92 | 32.15 | 30.26 |
| | FLAP | 37.02 | 41.13 | 26.29 | 32.96 | 29.43 | 33.09 | 32.51 |
| | Týr-the-Pruner | **42.62** | **49.45** | **33.68** | **39.71** | **39.99** | **38.68** | **40.24** |

Table 16: 0-shot acc (%) on OpenBookQA.

| Sparsity | Method | LLaMA-2 | | LLaMA-3.x | | | Mistral | |
|---|---|---|---|---|---|---|---|---|
| | | 7B | 13B | 2-3B | 0-8B | 1-8B | 7B-v0.3 | Nemo |
| 0% | N/A | 31.40 | 35.20 | 31.00 | 34.80 | 33.20 | 33.40 | 36.40 |
| 12.5% | ShortGPT | 28.20 | 33.20 | 26.60 | 33.00 | 30.80 | 27.00 | 31.60 |
| | LaCO+ | 30.00 | 30.20 | 25.60 | 30.80 | 31.20 | 25.60 | 28.80 |
| | SliceGPT | **32.00** | 34.00 | 27.20 | 28.60 | 26.60 | 27.20 | 26.40 |
| | Wanda-sp | 31.60 | 32.00 | 15.40 | 13.20 | 22.80 | 28.00 | 28.60 |
| | LLM-Pruner | 28.40 | 34.40 | 24.60 | 27.20 | 26.40 | 26.80 | 31.00 |
| | ZipLM | 31.60 | 34.60 | 27.50 | 25.80 | 26.60 | **34.20** | 32.80 |
| | OSSCAR | 31.20 | **35.80** | 27.00 | 25.40 | 26.40 | 32.60 | 31.80 |
| | FLAP | 29.20 | 32.40 | 27.60 | 30.60 | 30.40 | 33.40 | 31.60 |
| | Týr-the-Pruner | 31.20 | **35.80** | **29.20** | **33.40** | **34.60** | **34.20** | **34.80** |
| 25% | ShortGPT | 23.40 | 27.00 | 23.20 | 19.60 | 18.40 | 20.40 | 23.00 |
| | LaCO+ | 25.20 | 25.20 | 22.40 | 20.60 | 20.00 | 23.40 | 22.20 |
| | SliceGPT | 25.00 | 30.40 | 23.00 | 24.40 | 22.60 | 23.00 | 23.00 |
| | Wanda-sp | 29.20 | 17.80 | 12.40 | 14.20 | 13.40 | 26.60 | 21.20 |
| | LLM-Pruner | 21.00 | 28.80 | 15.40 | 19.80 | 18.00 | 20.40 | 21.00 |
| | ZipLM | 31.40 | **34.80** | 17.60 | 24.40 | 18.80 | 29.20 | 19.20 |
| | OSSCAR | 31.40 | 34.20 | 13.00 | 20.20 | 21.60 | 24.40 | 21.00 |
| | FLAP | 27.40 | 29.80 | 24.60 | 26.60 | 28.40 | 29.80 | 28.00 |
| | Týr-the-Pruner | 31.60 | 34.20 | 28.20 | 34.00 | 31.80 | 33.40 | 31.80 |
| 37.5% | ShortGPT | 21.60 | 22.00 | 18.80 | 18.80 | 18.40 | 17.20 | 17.20 |
| | LaCO+ | 17.00 | 21.40 | 17.00 | 17.80 | 16.60 | 17.00 | 19.80 |
| | SliceGPT | 19.80 | 27.20 | 17.60 | 17.00 | 15.00 | 17.00 | 17.40 |
| | Wanda-sp | 17.60 | 13.40 | 11.80 | 12.60 | 11.80 | 14.80 | 12.60 |
| | LLM-Pruner | 12.80 | 17.80 | 12.20 | 12.80 | 13.20 | 14.80 | 13.00 |
| | ZipLM | 25.60 | 27.00 | 13.80 | 14.40 | 13.20 | 21.80 | 14.60 |
| | OSSCAR | 25.20 | 26.40 | 14.20 | 15.20 | 14.80 | 21.60 | 21.00 |
| | FLAP | 24.20 | 27.20 | 21.60 | 22.80 | 23.40 | 24.00 | 25.80 |
| | Týr-the-Pruner | 31.00 | 32.40 | 26.00 | 29.80 | 30.00 | 26.20 | 29.20 |
| 50% | ShortGPT | 16.00 | 17.80 | 18.20 | 16.80 | 17.00 | 15.40 | 14.00 |
| | LaCO+ | 16.20 | 18.00 | 16.40 | 14.00 | 14.00 | 16.60 | 16.00 |
| | SliceGPT | 16.60 | 22.00 | 14.20 | 14.00 | 12.80 | 14.80 | 15.00 |
| | Wanda-sp | 12.20 | 11.80 | 13.00 | 13.60 | 13.40 | 13.60 | 13.80 |
| | LLM-Pruner | 12.60 | 12.00 | 12.40 | 13.40 | 13.40 | 14.40 | 14.60 |
| | ZipLM | 14.00 | 19.60 | 12.40 | 13.80 | 11.60 | 17.40 | 13.20 |
| | OSSCAR | 17.00 | 20.00 | 11.80 | 11.60 | 10.60 | 16.60 | 14.40 |
| | FLAP | 21.20 | 25.80 | 13.20 | 21.40 | 16.80 | 21.40 | 21.00 |
| | Týr-the-Pruner | 27.20 | 30.40 | 20.40 | 26.60 | 24.80 | 22.80 | 26.20 |

Table 17: 0-shot acc (%) on RTE.

| Sparsity | Method | LLaMA-2 | | LLaMA-3.x | | | Mistral | |
|---|---|---|---|---|---|---|---|---|
| | | 7B | 13B | 2-3B | 0-8B | 1-8B | 7B-v0.3 | Nemo |
| 0% | N/A | 62.82 | 65.34 | 54.87 | 67.87 | 71.12 | 68.95 | 64.26 |
| 12.5% | ShortGPT | 55.96 | 63.90 | 55.96 | 57.04 | 62.82 | **70.04** | 57.40 |
| | LaCO+ | 63.54 | 58.48 | 57.40 | **68.23** | 70.40 | 65.34 | **64.26** |
| | SliceGPT | 64.26 | 58.84 | 58.48 | 64.62 | 63.90 | 66.06 | 57.40 |
| | Wanda-sp | 58.48 | 64.98 | 46.93 | 57.04 | 57.76 | 57.76 | 62.82 |
| | LLM-Pruner | 62.82 | 61.73 | 48.38 | 56.68 | 61.01 | 65.34 | 55.60 |
| | ZipLM | 61.37 | 63.18 | 50.18 | 66.06 | 60.65 | 68.23 | 61.37 |
| | OSSCAR | 58.84 | 61.73 | 56.32 | 64.62 | 63.18 | 69.31 | 59.57 |
| | FLAP | 57.76 | 59.21 | 50.54 | 52.35 | 55.96 | 67.51 | 56.68 |
| | Týr-the-Pruner | 66.06 | 67.15 | 56.68 | 66.79 | 71.84 | 68.95 | 62.82 |
| 25% | ShortGPT | 57.76 | 59.57 | 48.38 | 62.82 | **63.90** | 64.98 | **63.54** |
| | LaCO+ | 53.79 | 62.45 | 57.04 | 63.90 | 58.12 | 64.26 | 58.12 |
| | SliceGPT | 55.96 | 66.79 | 59.21 | 58.12 | 57.40 | 57.40 | 52.71 |
| | Wanda-sp | 48.38 | 52.71 | 52.71 | 52.71 | 53.43 | 53.79 | 53.43 |
| | LLM-Pruner | 56.68 | 50.54 | 52.35 | 52.35 | 53.07 | 55.23 | 53.07 |
| | ZipLM | 55.60 | 68.23 | 55.23 | 50.18 | 51.99 | **68.95** | 53.43 |
| | OSSCAR | 52.35 | 67.51 | 48.74 | 54.51 | 50.54 | 63.18 | 53.79 |
| | FLAP | **62.82** | 64.26 | 53.43 | 50.90 | 52.71 | 63.90 | 49.46 |
| | Týr-the-Pruner | 62.09 | **69.31** | 59.57 | 63.90 | 63.18 | 65.34 | 59.57 |
| 37.5% | ShortGPT | 62.09 | 52.35 | **58.48** | 50.54 | 53.79 | 49.82 | 53.79 |
| | LaCO+ | 57.40 | 55.60 | 57.04 | 54.87 | 58.84 | **61.37** | 54.15 |
| | SliceGPT | 53.43 | 58.48 | 53.07 | 52.71 | 53.43 | 55.23 | 52.71 |
| | Wanda-sp | 48.38 | 52.71 | 54.51 | 46.57 | 50.54 | 53.07 | 49.46 |
| | LLM-Pruner | 52.71 | 52.71 | 52.71 | 52.71 | 52.71 | 51.26 | 52.71 |
| | ZipLM | 58.12 | 64.26 | 52.71 | 50.54 | 53.79 | 54.15 | 52.35 |
| | OSSCAR | 51.62 | 63.18 | 49.82 | 52.71 | 50.90 | 60.29 | 52.71 |
| | FLAP | 48.38 | 54.51 | 46.57 | 51.26 | 55.60 | 53.79 | **55.96** |
| | Týr-the-Pruner | 61.37 | **65.70** | 55.96 | 60.29 | 58.84 | 58.84 | 54.15 |
| 50% | ShortGPT | 51.26 | 51.62 | 51.99 | **60.65** | 51.62 | 51.26 | 50.54 |
| | LaCO+ | 51.62 | **61.01** | **57.40** | 51.62 | 45.13 | 53.43 | 49.82 |
| | SliceGPT | 53.43 | 52.71 | 53.07 | 53.43 | 55.96 | 53.07 | 52.71 |
| | Wanda-sp | 53.07 | 52.71 | 54.51 | 53.07 | 51.62 | 52.71 | 49.46 |
| | LLM-Pruner | 53.07 | 52.71 | 52.71 | 52.71 | 52.35 | 52.71 | 53.07 |
| | ZipLM | 52.71 | 52.71 | 53.07 | 52.35 | 52.71 | 52.71 | 51.26 |
| | OSSCAR | 53.43 | 52.71 | 50.90 | 47.65 | 51.26 | 53.79 | 55.96 |
| | FLAP | 45.49 | 58.48 | 51.62 | 53.07 | 52.71 | 52.71 | 55.23 |
| | Týr-the-Pruner | 55.96 | 59.93 | 53.43 | 58.84 | 58.12 | 53.79 | 60.65 |

Table 18: 0-shot acc (%) on WinoGrande.

| Sparsity | Method | LLaMA-2 | | LLaMA-3.x | | | Mistral | |
|---|---|---|---|---|---|---|---|---|
| | | 7B | 13B | 2-3B | 0-8B | 1-8B | 7B-v0.3 | Nemo |
| 0% | N/A | 69.06 | 72.22 | 69.06 | 73.01 | 73.56 | 73.64 | 73.64 |
| 12.5% | ShortGPT | 68.98 | 71.03 | 67.64 | 71.67 | 70.09 | 70.24 | **74.11** |
| | LaCO+ | 68.35 | 70.96 | **69.61** | 73.16 | **73.16** | 71.35 | 74.03 |
| | SliceGPT | 67.32 | 70.48 | 60.54 | 67.17 | 66.61 | 70.64 | 64.01 |
| | Wanda-sp | 67.64 | 70.72 | 52.01 | 51.38 | 58.80 | 66.54 | 62.27 |
| | LLM-Pruner | 64.25 | 69.30 | 60.93 | 65.35 | 65.51 | 66.06 | 68.51 |
| | ZipLM | **70.48** | 72.69 | 60.77 | 68.75 | 66.93 | 72.53 | 65.98 |
| | OSSCAR | 69.46 | 73.40 | 60.30 | 68.19 | 68.27 | 71.59 | 65.11 |
| | FLAP | 68.03 | 70.32 | 62.19 | 69.53 | 70.40 | 70.24 | 68.43 |
| | Týr-the-Pruner | 70.09 | **73.85** | 67.40 | **73.24** | 72.53 | 73.40 | 72.69 |
| 25% | ShortGPT | 65.67 | 70.96 | 61.40 | 53.99 | 55.17 | 67.25 | 63.14 |
| | LaCO+ | 64.25 | 69.77 | 63.14 | 67.32 | 65.27 | 67.80 | **71.59** |
| | SliceGPT | 65.27 | 70.40 | 57.62 | 62.67 | 59.75 | 61.48 | 58.25 |
| | Wanda-sp | 63.69 | 54.30 | 51.78 | 48.78 | 49.57 | 60.62 | 57.06 |
| | LLM-Pruner | 57.70 | 61.56 | 52.57 | 55.41 | 55.17 | 56.20 | 57.62 |
| | ZipLM | 67.96 | **72.38** | 51.85 | 58.88 | 58.25 | 66.93 | 54.70 |
| | OSSCAR | 68.11 | 70.56 | 52.72 | 56.91 | 56.75 | 64.33 | 55.41 |
| | FLAP | 64.72 | 68.03 | 57.06 | 62.75 | 63.46 | 64.72 | 64.64 |
| | Týr-the-Pruner | 68.51 | 72.06 | **64.01** | **71.11** | 71.11 | 71.11 | 70.01 |
| 37.5% | ShortGPT | 60.30 | 65.98 | **61.56** | 54.54 | 55.09 | 59.19 | 53.99 |
| | LaCO+ | 59.04 | 65.82 | 58.01 | 60.38 | 61.25 | 59.43 | 55.49 |
| | SliceGPT | 46.73 | 65.19 | 55.09 | 54.78 | 53.04 | 55.80 | 54.85 |
| | Wanda-sp | 49.49 | 49.57 | 48.38 | 51.30 | 49.49 | 49.33 | 51.38 |
| | LLM-Pruner | 51.07 | 52.57 | 49.96 | 51.07 | 50.83 | 51.78 | 50.36 |
| | ZipLM | 60.46 | 63.46 | 50.36 | 53.83 | 54.38 | 57.38 | 51.14 |
| | OSSCAR | 61.64 | 63.14 | 50.04 | 55.09 | 54.46 | 57.38 | 53.83 |
| | FLAP | 61.72 | 64.96 | 52.57 | 57.85 | 58.17 | 57.38 | 56.20 |
| | Týr-the-Pruner | 66.93 | 72.06 | 60.22 | 66.54 | 66.54 | 64.17 | 65.27 |
| 50% | ShortGPT | 56.20 | 61.80 | 51.46 | 54.70 | 54.14 | 53.12 | 52.64 |
| | LaCO+ | 51.38 | 59.19 | 51.22 | 53.75 | 52.17 | 51.30 | 50.12 |
| | SliceGPT | 54.62 | 56.99 | 51.30 | 50.99 | 49.88 | 51.38 | 49.33 |
| | Wanda-sp | 49.17 | 50.04 | 51.07 | 47.43 | 51.22 | 49.88 | 48.07 |
| | LLM-Pruner | 50.43 | 49.88 | 51.07 | 50.12 | 49.64 | 50.20 | 49.80 |
| | ZipLM | 48.54 | 56.91 | 51.54 | 49.17 | 52.49 | 51.93 | 50.59 |
| | OSSCAR | 51.54 | 52.80 | 51.70 | 48.86 | 51.14 | 52.49 | 50.43 |
| | FLAP | 56.51 | 61.72 | 50.51 | 52.80 | 54.14 | 52.88 | 52.09 |
| | Týr-the-Pruner | 62.12 | 70.09 | 53.28 | 60.30 | 61.80 | 59.43 | 59.04 |

Table 19: 5-shot acc (%) on MMLU.

| Sparsity | Method | LLaMA-2 | | LLaMA-3.x | | | Mistral | |
|---|---|---|---|---|---|---|---|---|
| | | 7B | 13B | 2-3B | 0-8B | 1-8B | 7B-v0.3 | Nemo |
| 0% | N/A | 45.84 | 55.06 | 65.27 | 56.17 | 65.20 | 62.18 | 68.83 |
| 12.5% | ShortGPT | **46.28** | 54.16 | 56.33 | **55.90** | 62.14 | 61.44 | **68.10** |
| | LaCO+ | 45.34 | **54.71** | 58.70 | 53.10 | **63.54** | **61.69** | 67.23 |
| | SliceGPT | 42.88 | 53.43 | 55.85 | 47.34 | 53.22 | 58.38 | 64.58 |
| | Wanda-sp | 39.99 | 44.64 | 25.78 | 26.69 | 43.26 | 56.42 | 58.22 |
| | LLM-Pruner | 37.40 | 50.51 | 48.71 | 39.30 | 50.26 | 54.09 | 54.59 |
| | ZipLM | 41.33 | 53.57 | 58.38 | 48.31 | 58.95 | 58.62 | 63.20 |
| | OSSCAR | 40.02 | 53.25 | 58.08 | 47.31 | 58.45 | 58.71 | 63.83 |
| | FLAP | 40.60 | 47.66 | 49.86 | 41.90 | 52.29 | 55.82 | 52.98 |
| | Týr-the-Pruner | 44.07 | 54.61 | **59.50** | 49.32 | 59.66 | 59.11 | 64.66 |
| 25% | ShortGPT | 37.38 | 48.00 | 36.10 | 34.30 | 34.88 | **59.66** | 50.66 |
| | LaCO+ | **43.63** | **53.82** | 39.36 | 35.78 | **59.68** | 55.28 | **62.63** |
| | SliceGPT | 40.20 | 51.21 | 36.46 | 35.02 | 32.30 | 45.76 | 55.35 |
| | Wanda-sp | 29.40 | 29.53 | 24.47 | 24.07 | 25.78 | 37.15 | 25.25 |
| | LLM-Pruner | 27.37 | 32.58 | 30.95 | 28.49 | 27.47 | 33.72 | 32.60 |
| | ZipLM | 32.79 | 45.99 | 41.67 | 24.79 | 42.25 | 51.22 | 51.12 |
| | OSSCAR | 31.04 | 46.28 | 44.89 | 24.89 | 43.06 | 49.51 | 51.18 |
| | FLAP | 30.57 | 42.66 | 35.96 | 34.47 | 39.18 | 47.99 | 31.09 |
| | Týr-the-Pruner | 34.90 | 52.18 | **52.12** | **42.66** | 51.22 | 52.17 | 57.68 |
| 37.5% | ShortGPT | **39.40** | 48.03 | 36.35 | 25.16 | 30.64 | 23.90 | **50.72** |
| | LaCO+ | 38.94 | **52.76** | 28.57 | 27.54 | 27.13 | 27.50 | 39.10 |
| | SliceGPT | 35.04 | 46.98 | 27.21 | **29.13** | 25.17 | 31.33 | 39.03 |
| | Wanda-sp | 25.79 | 24.05 | 23.87 | 23.44 | 25.62 | 26.61 | 22.99 |
| | LLM-Pruner | 24.65 | 25.50 | 24.38 | 24.63 | 26.88 | 25.97 | 25.79 |
| | ZipLM | 30.85 | 43.82 | 31.82 | 24.76 | 33.97 | 38.48 | 34.54 |
| | OSSCAR | 28.98 | 42.65 | 34.01 | 25.05 | 33.98 | 35.42 | 37.06 |
| | FLAP | 26.17 | 36.58 | 29.24 | 27.89 | 30.17 | 37.64 | 27.59 |
| | Týr-the-Pruner | 31.56 | 44.72 | **43.78** | 24.92 | **41.43** | **42.96** | 47.63 |
| 50% | ShortGPT | 25.77 | 23.97 | 23.00 | 24.24 | 22.97 | 23.37 | 32.22 |
| | LaCO+ | 23.81 | 40.19 | 25.95 | 25.48 | 26.08 | 24.47 | 25.35 |
| | SliceGPT | **29.37** | 38.94 | 24.93 | 25.54 | 25.17 | 26.34 | 28.24 |
| | Wanda-sp | 24.55 | 25.97 | 25.72 | 24.21 | 25.76 | 25.13 | 23.72 |
| | LLM-Pruner | 25.81 | 24.63 | 25.24 | 23.31 | 24.70 | 25.89 | 25.17 |
| | ZipLM | 25.70 | 29.40 | 25.35 | 24.87 | 26.16 | 27.92 | 28.46 |
| | OSSCAR | 25.35 | 31.36 | 25.71 | 25.60 | 26.33 | 25.27 | 27.97 |
| | FLAP | 23.87 | 29.40 | 23.49 | 23.40 | 23.18 | 25.07 | 24.23 |
| | Týr-the-Pruner | 26.06 | **40.29** | **30.46** | **26.46** | 33.76 | 33.51 | 33.34 |

