# OpenReview forum: "Týr-the-Pruner: Structural Pruning LLMs via Global Sparsity Distribution Optimization"
_NeurIPS.cc/2025/Conference — NeurIPS 2025 poster_

### Official Review · Reviewer_DF7P · 2025-06-14

**Clarity:** 3
**Significance:** 2
**Originality:** 3
**Rating:** 5
**Confidence:** 4

**Summary:**

The paper introduces Týr-the-Pruner, a global structural pruning framework for LLMs that optimizes sparsity distribution through an iterative prune-and-search approach. It achieves state-of-the-art performance with significant parameter reduction while maintaining high accuracy.

**Questions:**

1.	In Equation (3), the Taylor expansion of the error function $E=\left \| \mathbf{XW-X\widehat{W} } \right \| _2^{2}$ is expressed in terms of derivatives with respect to $\mathbf W$. However, $\mathbf W$ is fixed and the actual variable is either $\delta \mathbf W$ or equivalently $\widehat{\mathbf W}$. Could the authors clarify this?
2.	Equation (6) introduces two key hyperparameters, $\alpha_l$ and $\beta$. How sensitive is the final performance to different choices of $\alpha_l$ and $\beta$? Can the authors provide empirical results or practical guidance on setting these parameters?
3.	How does the search cost compare quantitatively to existing structured pruning methods such as LLM-Pruner and SliceGPT?
4.	Could the approach be integrated with other compression methods like quantization or unstructured pruning? If so, have the authors conducted experiments or analyses to evaluate the compatibility or additive effects of such combinations?

**Ethical Concerns:**

["NO or VERY MINOR ethics concerns only"]

**Final Justification:**

The author provides more detailed hyperparameter analysis and comparison on the running cost. Everything is clear.

**Limitations:**

Limitations are discussed. It would be better if the authors could optimize search time cost

**Quality:**

3

**Strengths And Weaknesses:**

Strengths:
1. The method overcomes the limitations of conventional two-stage pruning pipelines by integrating a locally pruned supernet with a global evolutionary search that is guided by expected error accumulation.
2. The proposed method is extensively evaluated on LLMs ranging from 7B to 70B parameters and achieves up to 97% of original accuracy even with 50% sparsity. It outperforms strong baselines in both perplexity and downstream QA tasks, demonstrating robustness and generalizability.

Weaknesses:
1. There are instances of carelessness in writing. For example, in line 71, the phrase “search the optimal sparsity distribution” is used, but the final letter ‘n’ in “distribution” is not bolded consistently with the rest of the word.
2. The search process incurs significantly higher computational cost compared to other structured pruning approaches such as LLM-Pruner and SliceGPT. Although the authors acknowledge this limitation and mention plans to improve efficiency in future work, the current method may be less practical in resource-constrained environments due to the heavy search overhead.

---

> ### Author Rebuttal · Authors · 2025-07-31
>
> Dear Reviewer DF7P:
>
> We greatly appreciate your insightful and constructive feedback. Please find our point-by-point responses to your concerns and questions below.
>
> ---
>
> **Weakness 1**: "There are instances of carelessness in writing. For example, in line 71, the phrase “search the optimal sparsity distribution” is used, but the final letter ‘n’ in “distribution” is not bolded consistently with the rest of the word."
>
> **Response to Weakness 1**: Thank you for pointing this out. We will correct the formatting inconsistency and carefully proofread the manuscript to eliminate similar and other typographical issues throughout the paper.
>
> ---
>
> **Weakness 2**: "The search process incurs significantly higher computational cost compared to other structured pruning approaches such as LLM-Pruner and SliceGPT. Although the authors acknowledge this limitation and mention plans to improve efficiency in future work, the current method may be less practical in resource-constrained environments due to the heavy search overhead."
>
> **Response to Weakness 2**: We acknowledge that there exists the cost of search in our method compared to some fast pruning approaches such as LLM-Pruner and SliceGPT. However, these methods often sacrifice accuracy and rely on costly post-pruning fine-tuning to recover performance. For example, as shown in *Table 2*, pruning 50% of LLaMA-3.1-70B results in SliceGPT and LLM-Pruner retaining only 62% and 45% of the dense model’s accuracy, respectively, whereas our method preserves up to 97%. For the former methods, substantial fine-tuning or knowledge distillation is typically required after pruning to restore accuracy—introducing additional computation and time overhead that can outweigh the initial pruning efficiency. Our method avoids gradient backpropagation during pruning (c.f., *line 158, Section 2.4*) and adopts an optimized supernet storage strategy (c.f., *Appendix A.4*), making the primary overhead come from search time rather than resource costs. The entire pruning process can be executed on only a single GPU for an 8B model, which ensures its applicability in resource-constrained environments. Overall, we believe that spending additional time to obtain a substantially higher-quality pruned model is a worthwhile trade-off, particularly in scenarios where inference efficiency and accuracy are both critical.
>
> ---
>
> **Question 1**: "In Equation (3), the Taylor expansion of the error function $E=|\mathbf{X} \mathbf{W}-\mathbf{X} \widehat{\mathbf{W}}|_2^2$ is expressed in terms of derivatives with respect to $\mathbf{W}$. However, $\mathbf{W}$ is fixed and the actual variable is either $\delta \mathbf{W}$ or equivalently $\hat{\mathbf{W}}$. Could the authors clarify this?"
>
> **Response to Question 1**: Thank you for the thoughtful question. While it is true that $\widehat{\mathbf{W}}$ is the optimization variable, our analysis adopts a perturbation-based perspective, where the focus is on how small deviations $\delta \mathbf{W} = \widehat{\mathbf{W}} - \mathbf{W}$ around a fixed reference point $\mathbf{W}$ influence the error function. The Taylor expansion is thus naturally formulated around $\mathbf{W}$. This approach is standard in model compression literature [1], where the goal is to assess the impact of weight perturbations. We will clarify this interpretation in the future version for better readability.
>
> ---
>
> **Question 2**: "Equation (6) introduces two key hyperparameters, $\alpha$ and $\beta$. How sensitive is the final performance to different choices of them? Can the authors provide empirical results or practical guidance on setting these parameters?"
>
> **Response to Question 2**: We express Equation (6) in a general form where the hyperparameter $\alpha_l$ controls the layer-wise alignment between the sparse and dense hidden representations, and $\beta$ governs the alignment between the output logits. In practice, we recommend setting the weights $\alpha_l$ uniformly across layers and normalizing them such that $\sum_l \alpha_l = \beta = 1$, following practices commonly used in typical knowledge distillation literature [2]. To assess the sensitivity to different supervision strategies, we conducted an ablation study comparing three configurations: (1) using only hidden-state alignment across layers (first, middle, and last layer), (2) using only output logits alignment, and (3) combining both types of supervision. We found that the combined supervision yields the best overall performance, indicating the complementary benefits of guiding both intermediate and final representations. However, layer-wise supervision requires intermediate activations checkpointing in memory. For practical purposes, we adopt logits-only supervision as it achieves a favorable trade-off between effectiveness and resource efficiency.
>
> **Table**: Ablation on Search Direction for Pruning 50% of Llama-3.1-8B
> | Search Direction| Perplexity | ARC-C | ARC-E | BoolQ | HellaSwag | OBQA | RTE | WinoGrande | MMLU | Average |
> |---|:---:|:---:|:---:|:---:|:---:|:---:|:---:|:---:|:---:|:---:|
> | No Search | 66.38 | 23.55 | 58.46 | 62.35 | 32.51 | 16.60 | 51.26 | 52.88 | 28.34 | 40.74 |
> | Hidden States Only | 37.86 | 30.97 | 62.05 | 63.16 | 33.40 | 21.80 | 53.79 | 57.38 | 29.99 | 44.07 |
> | Logits Only  | 30.89 | 31.83 | 65.36 | 66.64 | 39.99 | 24.80 | 58.12 | 61.80 | 33.76 | 47.79 |
> | Hidden States & Logits | 28.56 | 32.51 | 65.87 | 63.12 | 40.35 | 26.2 | 58.12 | 63.14 | 34.21 | 47.94 |
>
> ---
>
> **Question 3**: " How does the search cost compare quantitatively to existing structured pruning methods such as LLM-Pruner and SliceGPT?"
>
> **Response to Question 3**: Thank you for the helpful question. As shown in the Table below, LLM-Pruner, FLAP, and SliceGPT are representative post-training sparsity (PTS) methods that feature low optimization cost, making them attractive for fast deployment. However, they typically suffer from limited accuracy. In contrast, our method Týr-the-Pruner without search already outperforms all these PTS baselines in accuracy, while maintaining a similar or even lower optimization time. Furthermore, Týr-the-Pruner achieves the best overall performance across all benchmarks. Although it incurs a search cost, we believe this is a worthwhile trade-off: it produces a significantly more accurate and efficient sparse model without retraining, which is crucial in resource-constrained settings.
>
> **Table**: Cost and Performance Comparison when Pruning 50% Parameters of Llama-3.1-8B.
> | Method | Time cost | Wikitext2 | Arc-C | Arc-E | BoolQ | HellaSwag | OBQA | RTE | WinoGrande | MMLU | Average |
> |---|---|:---:|:---:|:---:|:---:|:---:|:---:|:---:|:---:|:---:|:---:|
> | LLM-Pruner | 18 minutes | 288.32 | 19.62 | 28.70 | 37.83 | 26.36 | 13.40 | 52.35 | 49.64 | 24.70 | 31.58 |
> | SliceGPT | 134 minutes | 353.21 | 21.50 | 41.62 | 38.56 | 29.07 | 12.80 | 55.96 | 49.88 | 25.17 | 34.32 |
> | FLAP | 22 minutes | 134.28 | 20.99 | 43.18 | 52.29 | 29.43 | 16.80 | 52.71 | 54.14 | 23.18 | 36.59 |
> | Týr-the-Pruner w/o search | 26 minutes | 66.38 | 23.55 | 58.46 | 62.35 | 32.51 | 16.60 | 51.26 | 52.88 | 28.34 | 40.74 |
> | Týr-the-Pruner | 612 minutes | 30.89 | 31.83 | 65.36 | 66.64 | 39.99 | 24.80 | 58.12 | 61.80 | 33.76 | 47.79 |
>
> ---
>
> **Question 4**: "Could the approach be integrated with other compression methods like quantization or unstructured pruning? If so, have the authors conducted experiments or analyses to evaluate the compatibility or additive effects of such combinations?"
>
> **Response to Question 4**: Thanks for the insightful question. Our method is indeed compatible with additional compression techniques. As shown in the Table below, we present results of applying quantization and unstructured sparsity on the 50% pruned Llama-3.1-8B model produced by Týr-the-Pruner. These results suggest that our pruned model is highly amenable to further compression: quantization maintains over 99% of the original pruned model's accuracy, while unstructured sparsity at 50% and 2:4 granularity achieves performance levels that are consistent with prior work (e.g., [SparseGPT: 91%@50%, ALPS: 83%@2:4] on 7B models). This compatibility indicates that Týr-the-Pruner can serve as a strong foundation in multi-stage compression pipelines, offering both high accuracy and robustness to further compression.
>
> **Table**: Integrated with quantization and unstructured pruning for 50% pruned Llama-3.1-8B.
> | Method | Wikitext2 | Arc-C | Arc-E | BoolQ | HellaSwag | OBQA | RTE | WinoGrande | MMLU | AVG |
> |---|:---:|:---:|:---:|:---:|:---:|:---:|:---:|:---:|:---:|:---:|
> | Týr-the-Pruner@FP16 | 30.89 | 31.83 | 65.36 | 66.64 | 39.99 | 24.80 | 58.12 | 61.80 | 33.76 | 47.79 |
> | Awq[3]@W4A16 | 34.79 | 31.06 | 64.65 | 66.09 | 39.34 | 24.60 | 60.77 | 60.77 | 31.43 | 47.34 (99.1%) |
> | SmoothQuant[4]@W8A8 | 31.31 | 31.23 | 64.94 | 65.50 | 40.18 | 25.80 | 58.84 | 61.80 | 32.45 | 47.59 (99.6%) |
> | RTN@FP8E4M3 | 31.05 | 31.31 | 65.28 | 66.45 | 40.17 | 24.60 | 58.84 | 61.48 | 32.28 | 47.55 (99.5%) |
> | SparseGPT[5]@50% | 47.05 | 27.99 | 59.39 | 65.32 | 37.38 | 23.60 | 53.07 | 61.48 | 28.32 | 44.57 (93.3%) |
> | ALPS[6]@M4N2 | 70.22 | 23.04 | 53.62 | 63.43 | 34.22 | 21.40 | 52.71 | 58.17 | 27.34 | 41.74 (87.3%) |
>
> ---
>
> [1] Optimal Brain Surgeon: Extensions and performance comparisons. 1993.
>
> [2] Sparse Fine-tuning for Inference Acceleration of Large Language Models. 2023.
>
> [3] AWQ: Activation-aware Weight Quantization for LLM Compression and Acceleration. 2024.
>
> [4] SmoothQuant: Accurate and Efficient Post-Training Quantization for Large Language Models. 2024.
>
> [5] SparseGPT: Massive Language Models Can Be Accurately Pruned in One-Shot. 2023.
>
> [6] ALPS: Improved Optimization for Highly Sparse One-Shot Pruning for Large Language Models. 2024.
>
> ---
>
> We sincerely appreciate your thoughtful feedback and support. If there are any points that would benefit from further explanation, we would be glad to elaborate.
>
> Sincerely,
>
> Authors 11393.

---

> > ### Comment · Reviewer_DF7P · 2025-08-01
> >
> > Good Work with Good Rebuttal, I will raise my score.

---

### Official Review · Reviewer_xbcw · 2025-06-16

**Clarity:** 4
**Significance:** 3
**Originality:** 3
**Rating:** 5
**Confidence:** 5

**Summary:**

This paper proposes a new global structural pruning approach called Tyr-the-Pruner. This method utilizes a supernet to determine the optimal sparsity distribution under a target overall sparsity ratio. An iterative prune-and-search strategy with coarse-to-fine sparsity granularity is used to reduce the search cost. The experimental results show that the proposed method can achieve state-of-the-art results.

**Questions:**

See weaknesses above.

**Ethical Concerns:**

["NO or VERY MINOR ethics concerns only"]

**Final Justification:**

The rebuttal has addressed all my concerns, so I suggest the acceptance of the paper.

**Limitations:**

yes.

**Paper Formatting Concerns:**

No.

**Quality:**

3

**Strengths And Weaknesses:**

Strengths:

1. This paper is technically sound and easy to understand.

2. The paper can discard 50% of channels without sacrificing much accuracy, which achieves state-of-the-art results.

3. The overall method shows some novelty.

Weaknesses:

1. Eq.4 is not very easy to understand. What is the intuition behind Eq.4? What does ~p mean?

2. How does random error accumulation and abandoned error accumulation do in Lines 149-150?

3. The authors use evolutionary search, which has many operations not optimized for GPU. What is the total search cost compared to sparsity-aware training methods and post-training sparsity methods?

4. Is this method comparable to or exceeding the sparsity-aware training methods?

5. You use 4 iterations during search. What if you use fewer or more iterations? Will that influence the search results and search time?

---

> ### Author Rebuttal · Authors · 2025-07-31
>
> Dear Reviewer xbcw:
>
> Thank you for your thoughtful and encouraging feedback. Below, we provide detailed responses to each of your comments and concerns.
>
> ---
>
> **Weakness 1**: "Eq.4 is not very easy to understand. What is the intuition behind Eq.4? What does ~p mean?"
>
> **Response to Weakness 1**: *Equation (4)* is derived based on the Taylor expansion for minimizing the pruning error (as shown in *Equations. (2-3)*). $W_p$ denotes the candidate channel selected for pruning, while $\delta W_{\sim p}$ represents the weight adjustment applied to the remaining unpruned channels ($\sim p$) to compensate for the error introduced by zeroing out $W_p$. For selecting which channel $p$ to prune, we prioritize those with both small gradient magnitude and low curvature (i.e., small entries in the Hessian, c.f., *Appendix A.1*). Intuitively, a small gradient implies that the weight contributes little to the loss, while a small Hessian entry indicates that the loss is relatively insensitive to perturbations in this weight. Therefore, pruning such a channel is expected to introduce minimal loss increase. For the remaining unpruned weights ($\sim p$), we derive an adjustment $\delta W_{\sim p}$ to compensate for the loss incurred by pruning $W_p$. This is formulated as a constrained optimization problem: minimizing the pruning-induced loss subject to the constraint that $W_p$ is set to zero. Using the method of Lagrange multipliers, we solve for the optimal $\delta W_{\sim p}$ in closed form, as shown in the right-hand expression of *Equation. (4)* and *Appendix A.1*. Overall and Intuitively, this formulation ensures that pruning and compensation jointly minimize the estimated error impact.
>
> ---
>
> **Weakness 2**: "How does random error accumulation and abandoned error accumulation do in Lines 149-150?"
>
> **Response to Weakness 2**: In the context of layer-wise pruning with calibration, error accumulation reflects how pruning decisions in earlier layers affect the inputs to later layers. Specifically, after pruning layer $l$, the input to layer $l+1$ becomes $X_{l+1} = W_l^{\text{pruned}} X_l$, thus propagating pruning-induced errors forward. Since our method explores multiple pruned candidates with different sparsity ratios per layer, we must decide how to accumulate errors. The abandoned strategy discards error accumulation entirely by using the dense model's original activations at every layer, ignoring any pruning-induced errors. In contrast, the random strategy selects one pruned candidate output at random and uses it as the input for the next layer's pruning, simulating a single-path forward pass but without fusion. Both are used as baselines to highlight the effectiveness of our proposed weighted accumulation in *Equation (5)*, which provides a more stable and realistic pruning signal.
>
> ---
>
> **Weakness 3**: "The authors use evolutionary search, which has many operations not optimized for GPU. What is the total search cost compared to sparsity-aware training methods and post-training sparsity methods?"
>
> **Weakness 4**: "Is this method comparable to or exceeding the sparsity-aware training methods?"
>
> **Response to Weakness 3 and 4**: To evaluate the trade-offs between efficiency and effectiveness, we conducted a comprehensive comparison targeting 50% sparsity on LLaMA-2-7B using four representative approaches: (1) *FLAP*, a previous SOTA post-training sparsity (PTS) method; (2) *Týr-the-Pruner without search*, which applies our local pruning but skips search; (3) *NutePrune [1]*, a training-aware sparsity method based on LoRA and multi-stage knowledge distillation; and (4) *Týr-the-Pruner* framework. As shown in the table below, PTS methods (FLAP and our non-search variant) complete in under 30 minutes, offering high efficiency but generally lower accuracy compared to training-aware/search methods. Nevertheless, our non-search variant consistently outperforms FLAP, demonstrating the strength of our local pruning strategy. While our full method incurs a higher search cost (~612 minutes), it significantly improves accuracy—surpassing both PTS and training-aware approaches. In particular, it outperforms the training-aware method NutePrune (649 minutes) in most tasks despite requiring no training. Moreover, our method does not rely on large-scale training data (requiring calibration samples only), making it more practical in scenarios with limited supervision. These results highlight that our approach strikes a strong balance between accuracy and efficiency while maintaining minimal data and computational requirements.
>
> **Table**: Comparison with post-training sparsity and training-aware sparsity methods for pruning 50% of Llama-2-7B’s parameters.
> | Method                    | Type                       | Optimization time cost | Wikitext2 | Arc-C | Arc-E | BoolQ | HellaSwag | OBQA  | RTE   | WinoGrande |
> |---------------------------|----------------------------|------------------------|-----------|-------|-------|-------|-----------|-------|-------|------------|
> | Dense                     | NA                         | 0 minutes              |    5.12   | 43.43 | 76.30 | 77.68 |   57.14   | 31.40 | 62.82 |    69.06   |
> | FLAP                      | post-training sparsity     | 22 minutes             |   25.49   | 29.10 | 47.01 | 58.50 |   37.02   | 21.20 | 45.49 |    56.51   |
> | Týr-the-Pruner w/o search | post-training sparsity     | 26 minutes             |   21.37   | 25.09 | 54.17 | 62.63 |   35.83   | 20.80 | 55.60 |    56.12   |
> | NutePrune                 | training-aware sparsity    | 649 minutes            |   12.90   | 35.41 | 62.46 | 63.64 |     -     |   -   |   -   |    61.25   |
> | Týr-the-Pruner            | iterative prune-and-search | 612 minutes            |   16.17   | 33.62 | 66.12 | 65.54 |   42.62   | 27.20 | 55.96 |    62.12   |
>
> ---
>
> **Weakness 5**: "You use 4 iterations during search. What if you use fewer or more iterations? Will that influence the search results and search time?"
>
> **Response to Weakness 5**: Thanks for the meaningful question. We conducted an ablation study by varying the number of search iterations from 0 (i.e., no search, equivalent to uniform pruning) to 5, while fixing the final iteration sparsity granularity to 1.5625% in each setting. As shown in the Table below, increasing the number of iterations consistently improves the performance of the searched sparsity pattern, with a smaller number of performance enhancements beyond 3–4 iterations. This suggests that 3–4 iterations are generally sufficient to reach near-optimal sparsity configurations. Each iteration takes approximately 150±10 minutes, and our default setting of 4 iterations offers a good trade-off between search cost and pruning quality. Intermediate results from each iteration in the 4-iteration setting can be found in *Table 8* in *Appendix A.6*.
>
> **Table**: Ablation Study of Search Iterations when pruning 50% parameters of Llama-3.1-8B.
> | Iterations | Wikitext2 | Arc-C | Arc-E | BoolQ | HellaSwag |  OBQA |  RTE  | WinoGrande |  MMLU |  AVG  |
> |------------|-----------|:-----:|:-----:|:-----:|:---------:|:-----:|:-----:|:----------:|:-----:|:-----:|
> | 0          |   66.38   | 23.55 | 58.46 | 62.35 |   32.51   | 16.60 | 51.26 |    52.88   | 28.34 | 40.74 |
> | 1          |   35.91   | 26.62 | 61.11 | 64.58 |   37.01   | 22.60 | 55.23 |    57.05   | 29.29 | 43.58 |
> | 2          |   37.16   | 31.66 | 65.61 | 63.61 |   37.79   | 23.00 | 58.84 |    56.83   | 30.65 | 46.00 |
> | 3          |   32.34   | 31.83 | 65.49 | 64.59 |   39.11   | 23.80 | 58.48 |    59.67   | 30.57 | 46.69 |
> | 4          |   30.89   | 31.83 | 65.36 | 66.64 |   39.99   | 24.80 | 58.12 |    61.80   | 33.76 | 47.79 |
> | 5          |   29.56   | 31.87 | 65.57 | 66.69 |   39.08   | 26.40 | 58.84 |    60.89   | 33.67 | 47.88 |
>
> ---
> [1] NutePrune: Efficient Progressive Pruning with Numerous Teachers for Large Language Models. 2024.
>
> ---
>
> We truly appreciate your valuable feedback and continued support. Should any aspect require further clarification, we would be happy to provide additional details.
>
> Sincerely,
>
> Authors 11393.

---

> > ### Comment · Reviewer_xbcw · 2025-08-04
> >
> > The rebuttal has addressed all my concerns， I would like to raise my evaluation score.

---

### Official Review · Reviewer_Atmm · 2025-06-22

**Clarity:** 2
**Significance:** 3
**Originality:** 3
**Rating:** 3
**Confidence:** 4

**Summary:**

This paper proposes a method for choosing the optimal sparsity values for different layers under the structured pruning setup. This is done by first pruning different layers to different sparsity levels, and then choosing the optimal sparsity allocation.

**Questions:**

See the comments above.

**Ethical Concerns:**

["NO or VERY MINOR ethics concerns only"]

**Final Justification:**

I think the lack of widely available inference implementation limits the applicability of the work. Given that this is not the first work to study non-uniform sparsity, I think the paper's contribution remains limited.

**Limitations:**

See the comments above.

**Quality:**

3

**Strengths And Weaknesses:**

I think this is an important problem to study, and I commend the authors for their work. The numerical experiments show the benefits of this type of sparsity allocation which is positive. However, I also have some concerns.

1- I believe the models that do not have uniform sparsity will need custom code, which will limit the application of this method in practice. Specially these custom models need to be implemented in serving engines such as vLLM and SGLang, which are used in practice for serving. This is missing from the paper.

2- The authors provide inference speed results for the models from their method, but how does this compare to a uniformly pruned model?   A non-uniform model will require invoking several different kernels on the GPU, which will create inference time overheads.

3- Are the accuracy numbers reported after distillation using (6)? How about the baselines? The consistency of the baselines need to be discussed.

4- What does (5) mean? Can you please provide some intuition?

5- (4) is very close to OBS/OSSCAR/ZipLM. Some discussion is needed here.

---

> ### Author Rebuttal · Authors · 2025-07-31
>
> Dear Reviewer Atmm:
>
> We are grateful for your careful and insightful comments. Please find our detailed responses to your observations and concerns below.
>
> ---
>
> **Weakness 1**: "I believe the models that do not have uniform sparsity will need custom code, which will limit the application of this method in practice. Specially these custom models need to be implemented in serving engines such as vLLM and SGLang, which are used in practice for serving. This is missing from the paper."
>
> **Weakness 2**: " The authors provide inference speed results for the models from their method, but how does this compare to a uniformly pruned model? A non-uniform model will require invoking several different kernels on the GPU, which will create inference time overheads."
>
> **Response to Weakness 1 and 2**: We appreciate your concerns regarding the practicality and efficiency of serving non-uniformly pruned LLMs. Our responses are as follows: **First**, our results demonstrate that non-uniform sparsity consistently outperforms uniform sparsity under the same compression ratio, achieving SOTA accuracy. This performance gap validates a hypothesis that weight importance is not evenly distributed across layers in LLMs. Thus, exploring non-uniform pruning is not merely a technical curiosity but a meaningful direction for understanding and optimizing LLMs.
>
> **Second**, while we acknowledge that current deployment frameworks like vLLM and SGLang primarily support uniform sparsity out of the box, it is the practical value of non-uniform sparsity that motivates extending these frameworks. A representative example is the recent push to support hybrid and non-uniform architectures in TensorRT-LLM, driven by large-scale Nemotron series models [1, 2]. Our work aligns with this trend and contributes to the ecosystem by identifying practical needs and performance benefits of such models.
>
> **Third**, we are actively working to extend vLLM to support non-uniform sparsity. This effort involves custom model architecture registration, tailored weight loading, and heterogeneous key-value (KV) cache management. Our support may be adopted by vLLM in the future. A preliminary result from this ongoing work is shown in the Table below.
>
> **Table**: Inference efficiency of uniform and non-uniform pruned Llama-3.1-8B. Benchmarks were conducted on a single AMD Instinct™ MI350 (288GB) accelerator using vLLM (extended to support non-uniform LLMs, with the temporary unified kv cache space allocation method) for inference, with input and output sequence lengths set to 256.
> | Sparsity           | TTFT (ms) | Prefill Speedup | Decoding Tpt. | Decoding Speedup |
> |--------------------|-----------|-----------------|---------------|------------------|
> |         0%         |   480.5   |      1.00x      |    7642.04    |       1.00x      |
> |     50% uniform    |   287.93  |      1.67x      |    11935.39   |       1.56x      |
> | 50% Týr-the-Pruner |   291.91  |      1.65x      |    11257.77   |       1.47x      |
>
> **Finally**, as shown in *Appendix A.5*, we compare the inference efficiency between uniformly and non-uniformly pruned models. The results show that inference efficiency are comparable in most cases. Uniform sparsity can offer slightly higher throughput only when the non-uniform sparsity distribution is highly imbalanced (i.e., large inter-layer sparsity variance), which causes multiple GEMM kernels (Including low-utilization "thin" MatMul) to be dispatched. Nonetheless, both pruned variants achieve substantial speedup over dense baselines, and the accuracy-efficiency tradeoff remains favorable for non-uniform models.
>
> ---
>
> **Weakness 3**: " Are the accuracy numbers reported after distillation using (6)? How about the baselines? The consistency of the baselines need to be discussed."
>
> **Response to Weakness 3**: We would like to clarify that our method does not involve knowledge distillation training. The formulation in Equation (6) is merely inspired by distillation, serving as a training-free scoring function to search for non-uniform sparsity patterns that better approximate the unpruned model’s outputs. All reported accuracy results are based on these searched architectures. Even without search, our method (Týr-the-Pruner with uniform sparsity) achieves competitive performance, as shown in the Table below (*Table 4 and Table 8* also provided relative detailed results). Additionally, to ensure fairness, **all baseline methods were re-evaluated using the same calibration dataset** as ours.
>
> **Table**: Týr-the-Pruner Performance with or without Search when pruning 50% parameters of Llama-3.1-8B.
> | Method                    | Wikitext2 | Arc-C | Arc-E | BoolQ | HellaSwag |  OBQA |  RTE  | WinoGrande |  MMLU |  AVG  |
> |---------------------------|-----------|:-----:|:-----:|:-----:|:---------:|:-----:|:-----:|:----------:|:-----:|:-----:|
> | LLM-Pruner                |   288.32  | 19.62 | 28.70 | 37.83 |   26.36   | 13.40 | 52.35 |    49.64   | 24.70 | 31.58 |
> | ZipLM                     |   366.34  | 20.48 | 28.28 | 57.43 |   26.42   | 11.60 | 52.71 |    52.49   | 26.16 | 34.45 |
> | FLAP                      |   134.28  | 20.99 | 43.18 | 52.29 |   29.43   | 16.80 | 52.71 |    54.14   | 23.18 | 36.59 |
> | Týr-the-Pruner w/o search |   66.38   | 23.55 | 58.46 | 62.35 |   32.51   | 16.60 | 51.26 |    52.88   | 28.34 | 40.74 |
> | Týr-the-Pruner            |   30.89   | 31.83 | 65.36 | 66.64 |   39.99   | 24.80 | 58.12 |    61.80   | 33.76 | 47.79 |
>
> ---
>
> **Weakness 4**: "What does (5) mean? Can you please provide some intuition?"
>
> **Response to Weakness 4**: Certainly. *Equation (5)* captures the idea of error accumulation during calibration-included layer-wise pruning. In this setting, pruning is performed layer by layer, where the input to each layer $X_l$ is affected by the pruning of the previous layer. After pruning layer $l$, the output becomes $X_{l+1}=W_l^{\text{pruned}}X_l$, which propagates errors forward. In our method, each weight generates multiple pruned versions with different sparsity levels. *Equation (5)* fuses the outputs from these variants using a **weighted combination**, where the weights are normalized sparsity scores—assigning **higher weight to lower-sparsity outputs**, which tend to be closer to the reliable original dense outputs. This fusion provides a more reliable input for pruning the next layer and better simulates cumulative pruning effects.
>
> ---
>
> **Weakness 5**: "(4) is very close to OBS/OSSCAR/ZipLM. Some discussion is needed here."
>
> **Response to Weakness 5**: We acknowledge that our method builds upon the same Taylor expansion principle as OBS, which has been widely adopted and validated in recent pruning works such as OSSCAR, ZipLM, SparseGPT, and LLM-Pruner. While large theoretical deviations are unlikely at this stage, our contribution lies in fully leveraging both first-order gradients and second-order Hessian information to guide fine-grained pruning and weight adjustments, pushing the performance boundary further. As shown in *Table 4*, our ablation study confirms the effectiveness of each component in our local pruning method. Most importantly, our core innovation is the iterative prune-and-search framework, within which the enhanced local pruning serves to build a more reliable supernet for sparsity distribution optimization.
>
> ---
>
> [1] Nemotron-H: A Family of Accurate and Efficient Hybrid Mamba-Transformer Models. 2025.
>
> [2] Llama-Nemotron: Efficient Reasoning Models. 2025.
>
> ---
>
> We sincerely thank you again for your thoughtful feedback and support. If there are any particular points that would benefit from further clarification, we would be glad to elaborate.
>
> Sincerely,
>
> Authors 11393.

---

> > ### Comment · Reviewer_Atmm · 2025-08-04
> >
> > Thank you for your response. I maintain my score, though I'm not strongly against accepting the paper.

---

### Official Review · Reviewer_qkjw · 2025-07-02

**Clarity:** 2
**Significance:** 3
**Originality:** 3
**Rating:** 5
**Confidence:** 4

**Summary:**

This paper introduces an efficient end-to-end search-based global structural pruning framework for LLMs. It proposes a method to construct a supernet by applying local pruning across various sparsity ratios to each layer of an LLM, aiming to identify the optimal sparsity distribution under a target overall sparsity ratio. The framework employs an iterative prune-and-search strategy with coarse-to-fine sparsity granularity to ensure efficient search convergence. The proposed method achieves state-of-the-art structural pruning results, retaining 97% of the dense model’s performance while removing 50% of Llama-3.1-70B’s parameters.

**Questions:**

Given that the matrix \hat{W} shares the same dimensionality as W, how can this sparser \hat{W} speed up inference? I have this concern because GPUs are optimized for tensor calculation and \hat{W}X and WX require the same number of computation operations.

**Ethical Concerns:**

["NO or VERY MINOR ethics concerns only"]

**Final Justification:**

My original score is already pretty positive,

**Limitations:**

yes

**Quality:**

3

**Strengths And Weaknesses:**

Strengths:
1. The paper is well-written and well-motivated.
2. The proposed method seems to be novel.
3. The experimental results presented are promising, showing the effectiveness and robustness of the proposed framework.

Weaknesses:
1. The proposed method requires an additional calibration dataset, however, some baselines do not rely on such a dataset, this makes the comparison unfair.
2. The method part could be polished to facilitate comprehension.

---

> ### Author Rebuttal · Authors · 2025-07-31
>
> Dear Reviewer qkjw:
>
> Thanks for your thoughtful and encouraging feedback. We are glad that you found our work valuable. In response to your concerns, we offer the following responses.
>
> ---
>
> **Weakness 1**: "The proposed method requires an additional calibration dataset, however, some baselines do not rely on such a dataset, this makes the comparison unfair."
>
> **Response to Weakness 1**: We understand the your concern regarding the fairness of comparisons due to the use of a calibration dataset. However, we would like to clarify that **(1)** all baselines in our main experiments were re-evaluated using the same calibration samples as our method, ensuring fair comparisons. This setup is described in *Section 3.1, line 225*. **(2)** For baselines in *Table 6 (Appendix A.3)*, whose official implementations were either released very recently or inaccessible to reproduce, we report results directly from their papers. Since these methods were already carefully tuned, we believe the comparison remains relatively fair. **(3)** Only a few methods—such as *magnitude pruning* and *PruneNet (a baseline in Table 6)*—are not calibration-dependent, but either perform poorly or are designed specifically in a calibration-free manner, making their inclusion reasonable for a comparison.
>
> ---
>
> **Weakness 2**: "The method part could be polished to facilitate comprehension."
>
> **Response to Weakness 2**: We agree that the method section could be made clearer to improve reader comprehension. In particular, we realize that adding an overview of the overall algorithmic process and providing richer illustrative examples would significantly enhance clarity. We will refine this part accordingly in future revisions.
>
> ---
>
> **Question**: "Given that the matrix $\widehat{\mathbf{W}}$ shares the same dimensionality as $\mathbf{W}$, how can this sparser $\widehat{\mathbf{W}}$ speed up inference? I have this concern because GPUs are optimized for tensor calculation and $\widehat{\mathbf{W}}\mathbf{X}$ and $\mathbf{W}\mathbf{X}$ require the same number of computation operations."
>
> **Response to Question**: Thank you for the insightful question. We believe the concern about pruning and its impact on actual inference speed is understandable but unnecessary in our case. In our method, the pruning process is initially represented via setting zeros, with weight dimensions kept unchanged during our iterative prune-and-search process (as shown in *Equations* and *Figure 2(b-c)*). After the algorithm is finalized, we perform **true structural compression** by removing the zeroed **rows and columns**—specifically, reducing the **output** dimensions of weights of *q*, *k*, *v*, *up*, and *gate*, and the **input** dimensions of weights of *o* and *down*. Other modules such as RMSNorm, Embeddings, and the LM-head remain unchanged. The resulting model, illustrated as *Figure 2(d)*, achieves genuine reductions in parameter count and FLOPs, yielding **practical speedup without relying on hardware-specific support**.
>
> You can view detailed information about the actual acceleration of the post-pruned model in *Table 3 (Subsection 3.2)* and *Figure 4 (Appendix A.5)*. Or refer to the following simple table to quickly summarize the post-pruning acceleration effect.
>
> **Table**: Inference efficiency of post-pruned LLMs with Týr-the-Pruner. Benchmarks were conducted on a single AMD Instinct™ MI250 accelerator using PyTorch (HipBlas) for LLM inference, with input and output sequence lengths set to 2048.
>
> | Model         | Sparsity | #Params | TTFT        | Decode Throughput |
> |---------------|----------|---------|-------------|-------------------|
> | Llama-3.1-8B  | 0%       | 8.0B    | 2.49 (1.00×) | 12.27 (1.00×)     |
> | Llama-3.1-8B  | 25%      | 6.1B    | 1.94 (1.28×) | 14.13 (1.15×)     |
> | Llama-3.1-8B  | 50%      | 4.3B    | 1.42 (1.75×) | 16.97 (1.38×)     |
>
> ---
>
> Please let us know if any specific aspect could be further elaborated. We thank you again for your constructive review and your support of our work.
>
> Sincerely,
>
> Authors 11393.

---

> > ### Comment · Reviewer_qkjw · 2025-08-05
> >
> > Thanks for the response, my original score reflects my opinion for this work.

---

### Decision · Program_Chairs · 2025-09-17

**Decision:**

Accept (poster)

**Comment:**

The paper presents an end‑to‑end, search‑based global structural pruning framework that builds a supernet of locally pruned sub‑models and efficiently optimizes the layer‑wise sparsity distribution under a target overall compression ratio. The method introduces an expectation‑error accumulation scheme for reliable supernet construction and an iterative prune‑and‑search strategy with coarse‑to‑fine granularity, which together enable state‑of‑the‑art results: \approx 97 % of dense performance is retained while removing 50 % of the parameters of Llama‑3.1‑70B (and comparable gains on smaller models). Extensive experiments support the approach and look promising. Overall, the contribution is technically solid, well‑validated, and of high relevance to the community working on efficient LLM deployment. The majority of strong, high‑confidence reviews support acceptance, and the single borderline reject raises only practical concerns that seem to be mostly addressed in the rebuttal. I concur and recommend acceptance.